*Report*

EMBO
Molecular Medicine

# IL-36 signaling as a drug target in Crohn's disease patients with *IL36RN* mutations

Julia Hecker [1], Christina Plattner [2], Camila A Cancino[1,3], Britt-Sabina Löscher[4], Judith Saurenbach[1], Marilena Letizia[1], Dietmar Rieder[2], TRR241 IBDome Consortium*, Inka Freise[1], Kristina Koop[5], Clemens Neufert[5], Désirée Kunkel[6], Zainab Al Khatim[1], Lars-Arne Schaafs[7], Anja Schütz [8], Christoph Becker[5], Raja Atreya[5], Zlatko Trajanoski [2], Andre Franke[4], Elena Sonnenberg[1], Ahmed N Hegazy[1,3], Britta Siegmund [1,9 ✉] & Carl Weidinger [1,9 ✉]

## Abstract

**The IL-36 signaling pathway has recently been identified as a key regulator of intestinal homeostasis and inflammation. However, the role of mutations in the IL-36R signaling pathway in the pathogenesis of inflammatory bowel disease remains unclear. We here identified four Crohn's disease patients with heterozygous missense mutations in the IL-36 receptor antagonist (*IL36RN*, IL-36RA). Experimental overexpression and functional assays demonstrated that two identified mutations resulted in reduced expression of IL-36RA. In-depth immune profiling of one *IL36RN*-mutated patient revealed an increased response of PBMCs to IL-36 stimulation and elevated serum levels of IL-36-regulated cytokines. Administration of the IL-36R-blocking antibody spesolimab to this patient resulted in a reduction of intestinal inflammation and alterations in immune cell composition and function. Our findings indicate that pathogenic *IL36RN* mutations may contribute to the pathogenesis of Crohn's disease in a subset of patients and that inhibiting IL-36 signaling could offer a personalized therapeutic approach for these patients.**

**Keywords** IL-36 Signaling; Crohn's Disease; Personalized Therapy; Genetics
**Subject Categories** Digestive System; Genetics, Gene Therapy & Genetic Disease; Immunology

## Introduction

Inflammatory bowel diseases (IBD) are defined by chronic inflammation of the gastrointestinal tract and are typically classified into the two main subtypes Crohn's disease (CD) and ulcerative colitis (UC). The etiology of IBD remains incompletely understood, but previous studies have shown that a complex interplay of environmental, genetic, microbial and immune factors is involved in the development of IBD. Genetic factors include risk loci identified by genome-wide association studies (GWAS) and rare mutations leading to severe and very early-onset monogenic IBD (Loddo and Romano, 2015). The study of genetic factors of IBD has provided insights into the pathways involved in the pathogenesis of IBD and contributed to the development of new therapies targeting these pathways (Neurath, 2019). However, despite these advances in IBD therapy, the remission rates for individual treatments do often not exceed 30–40% and some patients do not show satisfying long-lasting clinical responses with any available treatment option (Alsoud et al, 2021). Consequently, the identification of novel therapeutic targets and predictors of clinical response is of paramount importance to facilitate a personalized and efficacious treatment approach for IBD patients.

The IL-36R signaling pathway has recently been identified as a key regulator of intestinal homeostasis and tissue remodeling (Scheibe et al, 2017; Scheibe et al, 2019). The IL-36 cytokine family belongs to the IL-1 family and comprises three agonists IL-36α, IL-36β, and IL-36γ, and the antagonist, IL-36RA. The binding of IL-36α, IL-36β, and IL-36γ to the IL-36R leads to the activation of NFκB and the production of pro-inflammatory cytokines such as IL-6 and TNFα, whereas the binding of IL-36RA to the IL-36R inhibits IL-36R signaling (Towne et al, 2004). Studies in mice have indicated that IL-36R signaling possesses dual functions in IBD: In acute intestinal inflammation, IL-36R signaling appears to be important for wound healing (Scheibe et al, 2017). Conversely, in

[1]Department of Gastroenterology, Infectious Diseases and Rheumatology, Charité - Universitätsmedizin Berlin, corporate member of Freie Universität Berlin, Humboldt-Universität zu Berlin, Campus Benjamin Franklin, Berlin, Germany. [2]Biocenter, Institute of Bioinformatics, Medical University of Innsbruck, Innsbruck, Austria. [3]Deutsches Rheuma-Forschungszentrum, ein Institut der Leibniz-Gemeinschaft, Berlin, Germany. [4]Institute of Clinical Molecular Biology, Kiel University and University Medical Center, Kiel, Germany. [5]Department of Medicine 1, Friedrich-Alexander-Universität Erlangen-Nürnberg, Erlangen, Germany. [6]Berlin Institute of Health at Charité - Universitätsmedizin Berlin, Flow & Mass Cytometry Core Facility, Berlin, Germany. [7]Department of Radiology, Charité - Universitätsmedizin Berlin, Berlin, Germany. [8]Max-Delbrück-Center for Molecular Medicine in the Helmholtz Association (MDC), Berlin, Germany. [9]These authors contributed equally: Britta Siegmund, Carl Weidinger. *A list of authors and their affiliations appears at the end of the paper. ✉E-mail: britta.siegmund@charite.de; carl.weidinger@charite.de

chronic intestinal inflammation, it contributes to the development of inflammation and fibrosis (Scheibe et al, 2019). Furthermore, it has been demonstrated that the expression of IL-36 is upregulated in the mucosa of UC and CD patients and in stenotic areas of IBD patients (Boutet et al, 2016; Nishida et al, 2016; Scheibe et al, 2019). These data indicate that tight regulation of IL-36R signaling is required to maintain intestinal homeostasis and that defects in the IL-36R signaling pathway could result in the development of intestinal inflammation.

Defects in IL-36R signaling have been described in generalized pustular psoriasis (GPP), where heterozygous, compound heterozygous and homozygous mutations in the gene encoding IL-36RA (*IL36RN*) lead to the rare deficiency of interleukin thirty-six–receptor antagonist (DITRA) syndrome (Marrakchi et al, 2011; Tauber et al, 2016). However, until now the prevalence and role of mutations in the IL-36R signaling pathway in IBD remain elusive. We here show that *IL36RN* mutations are present in a subset of CD patients and that the blockade of IL-36 signaling may represent a personalized therapeutic approach for this rare subset of patients in case of therapeutic failure of other advanced therapies.

## Results

A 27-year-old Caucasian female presented herself at our department with a severe and therapy-refractory course of CD with ileal inflammation and fistulizing/penetrating disease (Montreal classification A2 L3 B3p). As shown in Fig. 1A, the patient was initially diagnosed with CD at the age of 18 and subsequently treated with all available therapeutic options for CD, including immunomodulators, biologics, and small molecules. Due to an inadequate clinical response to these therapies, the patient underwent multiple bowel resections, recurrent draining of pelvic abscesses and perianal fistulas, and even received an autologous stem cell transplantation (Fig. 1A; Appendix Table S1). However, none of these interventions resulted in a long-term clinical remission or a satisfying clinical response with adequate control of symptoms. To elucidate the underlying mechanisms responsible for the severe and therapy-refractory course of CD and to identify potential therapeutic targets, we performed whole-exome sequencing (WES) of peripheral blood mononuclear cells (PBMCs) of the patient. Thereby, we detected a heterozygous missense mutation in *IL36RN* (*IL36RN* S113L) (Fig. 1B ; Appendix Fig. S1), which had previously been described in patients with GPP (Onoufriadis et al, 2011). Notably, we did not observe any known mutations associated with the development of inflammatory bowel diseases.

To investigate the effect of the identified mutation on the expression and function of IL-36RA, we overexpressed the *IL36RN* S113L variant in HEK 293 T cells. In line with previous reports, we observed that the *IL36RN* S113L mutation results in a reduced expression of IL-36RA (Tauber et al, 2016) (Fig. 1C,D). Furthermore, we produced recombinant IL-36RA S113L and analyzed its ability to antagonize IL-36α stimulation. As shown in Fig. 1E, we observed that both WT IL-36RA and mutant IL-36RA S113L could antagonize IL-36α stimulation in a comparable manner, suggesting that the *IL36RN* S113L mutation primarily affects the expression of IL-36RA and not its function.

To assess the functional impact of the *IL36RN* S113L mutation on IL-36R regulation in the *IL36RN*-mutated patient (hereafter

referred to as the IL-36RA patient), we stimulated PBMCs of the IL-36RA patient with IL-36α in vitro and measured the production of the NFκB-induced cytokines TNFα, IL-6, and IL-8. We observed that PBMCs of the IL-36RA patient produced increased levels of TNFα, IL-6, and IL-8 compared to a healthy donor (HD), suggesting defects in the regulation of IL-36R signaling activity in the IL-36RA patient (Fig. 1F).

Given that IL-36R signaling has been demonstrated to influence the differentiation and functionality of immune cells (Carrier et al, 2011; Dietrich et al, 2016), we conducted a comprehensive analysis of PBMCs from the IL-36RA patient using mass cytometry and quantified serum levels of 13 inflammatory cytokines by cytometric bead array (CBA) (Appendix Figs. S2, S3, and S4). In comparison to HDs and CD patients, we observed elevated levels of IL-23 and IL-18 in the serum of the IL-36RA patient (Fig. 1G). Furthermore, we noted that PBMCs produced IL-18 and IL-23 upon IL-36α stimulation in vitro and that the fold change in cytokine production, particularly IL-23, was increased in PBMCs of the IL-36RA patient compared to an HD (Fig. EV1A,B). Consistent with the elevated IL-23 levels in the serum, we observed an increased abundance of Th17 cells in the PBMCs of the IL-36RA patient in comparison to healthy donors or non-mutated CD patients. In addition, the frequency of B cells was increased, while that of NK cells was reduced in PBMCs of the IL-36RA patient compared to the control groups (Fig. 1H). We also analyzed our CyTOF data to identify which cells produce pro-inflammatory cytokines in response to IL-36α stimulation and found that myeloid cells are the primary responders to IL-36α stimulation in the blood (Appendix Fig. S5).

To determine if blocking IL-36R signaling could reduce the production of pro-inflammatory cytokines observed in our patient, we stimulated PBMCs of the *IL36RN*-mutated patient with IL-36α in vitro in the presence or absence of the IL-36R blocking antibody spesolimab. As shown in Fig. EV1C, stimulation of cells with IL-36α resulted in an enhanced secretion of IL-6, TNFα, and IL-23, which could be significantly reduced by treatment with spesolimab. Thus, our data indicate that the identified mutation in the *IL36RN* gene results in a reduced expression of IL-36RA and consequently in overactivation of the IL-36R signaling pathway in our patient, suggesting that blockade of IL-36R signaling might represent an effective target for immunosuppression in the IL-36RA patient.

Given the association between *IL36RN* mutations and the development of GPP (Marrakchi et al, 2011; Onoufriadis et al, 2011) and the recent approval of the anti-IL-36R blocking antibody spesolimab for the treatment of GPP (Burden et al, 2023), we decided to treat our IL-36RA patient with spesolimab based on an out-of-scope application of spesolimab. The treatment regimen consisted of three cycles of cyclophosphamide, followed by monthly intravenous infusions of 1200 mg spesolimab and subcutaneous injections of certolizumab pegol (Fig. 2A). A combination therapy of spesolimab and the anti-TNFα antibody certolizumab pegol was used according to the phase 2 clinical IL-36R trial protocol (NCT04362254), investigating the efficacy and safety of spesolimab in CD patients in a combined administration with TNF-blockers (Ferrante et al, 2023). This approach resulted in a reduction in intestinal inflammation and partial clinical response, as evidenced by a decline in calprotectin levels and a reduction in the Simple Endoscopic Score for Crohn's diseases (SES-CD) at week 12 of spesolimab treatment (Fig. 2B,C; Appendix Table S2). Furthermore,

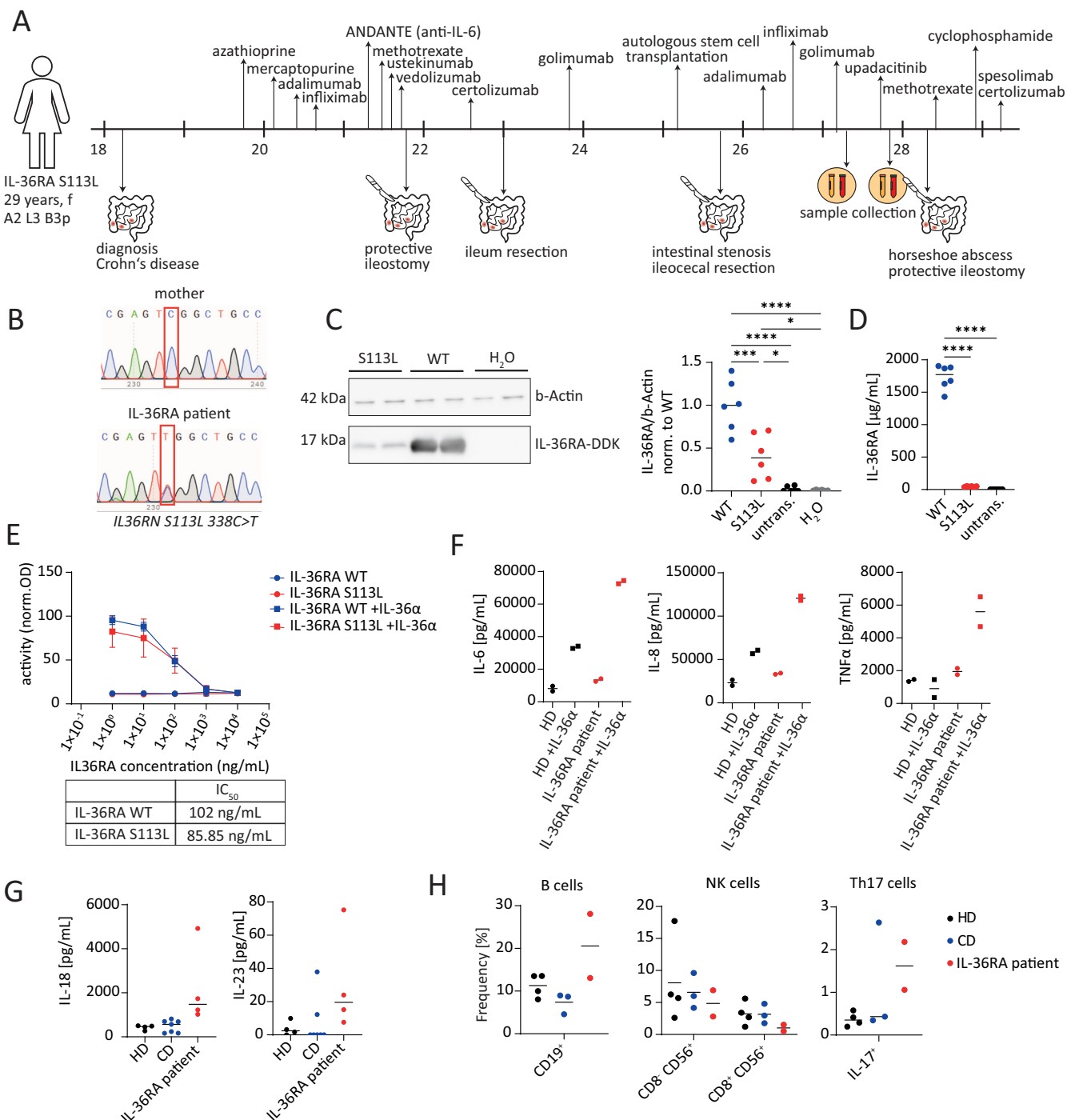

we also observed the complete healing of a large pelvic horseshoe abscess under the sequential treatment with cyclophosphamide and spesolimab/certolizumab pegol as assessed by MRI scans (Fig. EV2C; Appendix Fig. S6).

To investigate the molecular changes induced by spesolimab therapy, we collected PBMCs and serum samples before and during the therapy and analyzed them by mass cytometry and CBA, respectively (Appendix Figs. S7 and S8). Our findings revealed that the concentration of IL-18 and IL-23 in the serum, which had been

elevated prior to therapy, decreased during spesolimab treatment (Fig. 2D). From week 8 of treatment with spesolimab and certolizumab pegol, we observed an increase in the abundance of NK cells and a decrease in the abundance of myeloid cells in PBMCs of the IL-36RA patient (Fig. 2E,F). In addition, we detected a reduced frequency of IL-6$^+$, IL-8$^+$, and TNFα$^+$ myeloid cells in IL-36α-stimulated PBMCs at weeks 8 and week 12 of treatment compared to baseline (Fig. 2G). Furthermore, we performed bulk RNA sequencing of intestinal biopsies of the IL-36RA patient, as

**Figure 1. Identification of a pathogenic *IL36RN* mutation in a Crohn's disease patient.**

(A) Summary of the clinical history of the *IL36RN*-mutated patient (IL-36RA patient). (B) Sanger sequencing of EDTA blood samples from the IL-36RA patient and the patient's mother was conducted. The red square indicates the position of the mutation (c.338C) in the *IL36RN* gene. (C, D) HEK 293T cells were transfected with either *IL36RN* wild-type (WT), *IL36RN* S113L (S113L), water ($H_2O$), or were left untransfected (untrans.). The expression of IL-36RA was analyzed 48 h after transfection (C) in the supernatant by ELISA and (D) in cell lysates by Western blot analyses. Data represent three independent experiments with $n = 6$ per condition. The line in the plots indicates the median. (E) The activity of NFκB in HEK-Blue IL-36 cells that had been pre-incubated with different concentrations of IL-36RA WT or IL-36RA S113L and subsequently stimulated with IL-36α. The values were normalized to control samples stimulated with IL-36α only. The half maximal inhibitory concentration ($IC_{50}$) for IL-36RA WT and IL-36RA S113L is indicated in the table below the graph. The data represent four independent experiments. The data are represented as mean ± SD ($n = 8$). (F) Peripheral blood mononuclear cells (PBMCs) from the IL-36RA patient and one healthy donor (HD) were stimulated in vitro with IL-36α for 7 h or left unstimulated (unstim.). Subsequently, cytokine levels in the supernatant were analyzed. Duplicates represent technical replicates. The line in the plots indicates the median. (G) The serum cytokine levels of healthy donors (HDs), Crohn's disease patients (CD), and the IL-36RA patient at different time points are presented. The line in the plots indicates the median. HD: $n = 4$, CD: $n = 7$, IL-36RA patient: $n = 4$ (same patient, different time points). (H) PBMCs of the IL-36RA patient at two different time points, CD patients, and HDs were stimulated with phorbol 12-myristate 13-acetate (PMA)/ionomycin (Iono) or lipopolysaccharide (LPS) for 4 h or with IL-36α for 7 h and subsequently analyzed by mass cytometry. The frequency of NK cells and B cells in unstimulated samples and the frequency of Th17 cells in PMA/Iono-stimulated samples are presented. The line in the plots indicates the median. HD: $n = 4$, CD: $n = 3$, IL-36RA patient: $n = 2$ (same patient, different time points). The statistical significance in (C, D) was determined by one-way ANOVA with Tukey's multiple comparisons test. ****$P < 0.0001$, ***$P < 0.001$, **$P < 0.01$, *$P < 0.05$. Exact $P$ values for the statistical comparisons are shown in Appendix Table S3. Source data are available online for this figure.

well as FACS analyses of lamina propria mononuclear cells (LPMCs) isolated from intestinal biopsies obtained before and during spesolimab therapy. Bulk RNA sequencing analyses revealed that samples of the IL-36RA patient clustered distinctly from those of HDs and non-mutated CD patients. Differential gene expression analyses thereby showed a significant downregulation of several *HOX* family genes in the ileum and a significant upregulation of genes associated with antimicrobial responses and tissue repair such as *REG3A*, *REG1B*, and *DEFA5* in the colon of the IL-36RA patient, which could be further enhanced by treatment with spesolimab (Appendix Fig. S9). FACS analyses furthermore revealed an increased frequency of activated CD8+ CD38+ T cells in both LPMCs of the colon and ileum of the IL-36RA patient in comparison to LMPCs obtained from CD controls, which markedly declined during spesolimab treatment (Fig. EV2A,B). Taken together, these data indicate that spesolimab treatment reduced the inflammatory burden both systemically and, in the intestine, leading to a partial clinical response in the *IL36RN*-mutated patient.

Finally, we aimed to determine whether pathogenic *IL36RN* mutations are also present in other IBD patients. To this end, we searched in a previously generated WES dataset from the TRR241 IBDome consortium including 86 UC patients, 244 CD patients and 45 non-inflamed controls for patients with pathogenic mutations in *IL36RN*. Surprisingly, we identified three additional patients with heterozygous missense mutations in *IL36RN*. All three patients presented with CD, with a predominant manifestation in the terminal ileum and a disease onset in adulthood. However, only one patient carried the previously identified *IL36RN* S113L mutation, while the two other patients displayed different mutations (*IL36RN* P76L and *IL36RN* L133I) (Fig. 3A). The Genome Aggregation Database (gnomAD) (Karczewski et al, 2020) was employed to ascertain the prevalence of the identified mutations in the healthy population. The allele frequencies of the identified mutations were determined to be low in the healthy population. The frequencies were 0.0028 for *IL36RN* S113L, 0.000265 for *IL36RN* P76L, and no database entry was found for *IL36RN* L133I.

In accordance with our previous experiments with *IL36RN* S113L, we also investigated the impact of the *IL36RN* P76L and the *IL36RN* L133I mutations on the expression and function of

IL-36RA. Our findings indicated that *IL36RN* P76L resulted in a reduction in IL-36RA expression, similar to that observed for *IL36RN* S113L. In contrast, no differences in expression or function were found for *IL36RN* L133I (Fig. 3B–E).

## Discussion

In summary, we identified four CD patients with mutations in *IL36RN* and observed in one *IL36RN*-mutated therapy-refractory patient a reduction in intestinal inflammation upon anti-IL-36R therapy.

Given that *IL36RN* mutations have only been described in patients with GPP and other skin diseases, this study provides, to our knowledge, the first description of *IL36RN* mutations in IBD. The results of overexpression experiments demonstrated that the two identified mutations lead to a reduction in IL-36RA expression. Given that elevated IL-36R signaling is linked to intestinal inflammation and fibrosis (Scheibe et al, 2019), it can be hypothesized that the identified mutations might at least partially contribute to the pathogenesis of CD in those patients by leading to an increased IL-36R signaling activity due to a reduced expression of IL-36RA. Following this hypothesis, we observed an enhanced production of pro-inflammatory cytokines in response to IL-36α stimulation and an increased concentration of the IL-36-regulated cytokines IL-23 and IL-18 in the serum of an *IL36RN*-mutated patient. Additionally, in vitro treatment of PBMCs of the *IL36RN*-mutated patient with spesolimab reduced the production of pro-inflammatory cytokines, including IL-23, IL-6, and TNFα upon IL-36α stimulation. Our analyses further demonstrated that these cytokines are primarily produced by myeloid cells in the blood. However, it is important to consider that in tissue, other cell types, such as macrophages, epithelial cells, and fibroblasts, may also respond to IL-36 stimulation.

Furthermore, the patient showed a partial clinical response to treatment with the IL-36R-blocking antibody spesolimab in combination with certolizumab pegol. This finding provides additional evidence that IL-36R signaling is involved in the inflammatory response in this patient. In addition, our findings indicate that the therapy was associated with a reduction in IL-18 and IL-23 levels in the serum, as well as a decreased abundance of pro-inflammatory IL-6+, IL-8+, and TNFα+ monocytes in PBMCs.

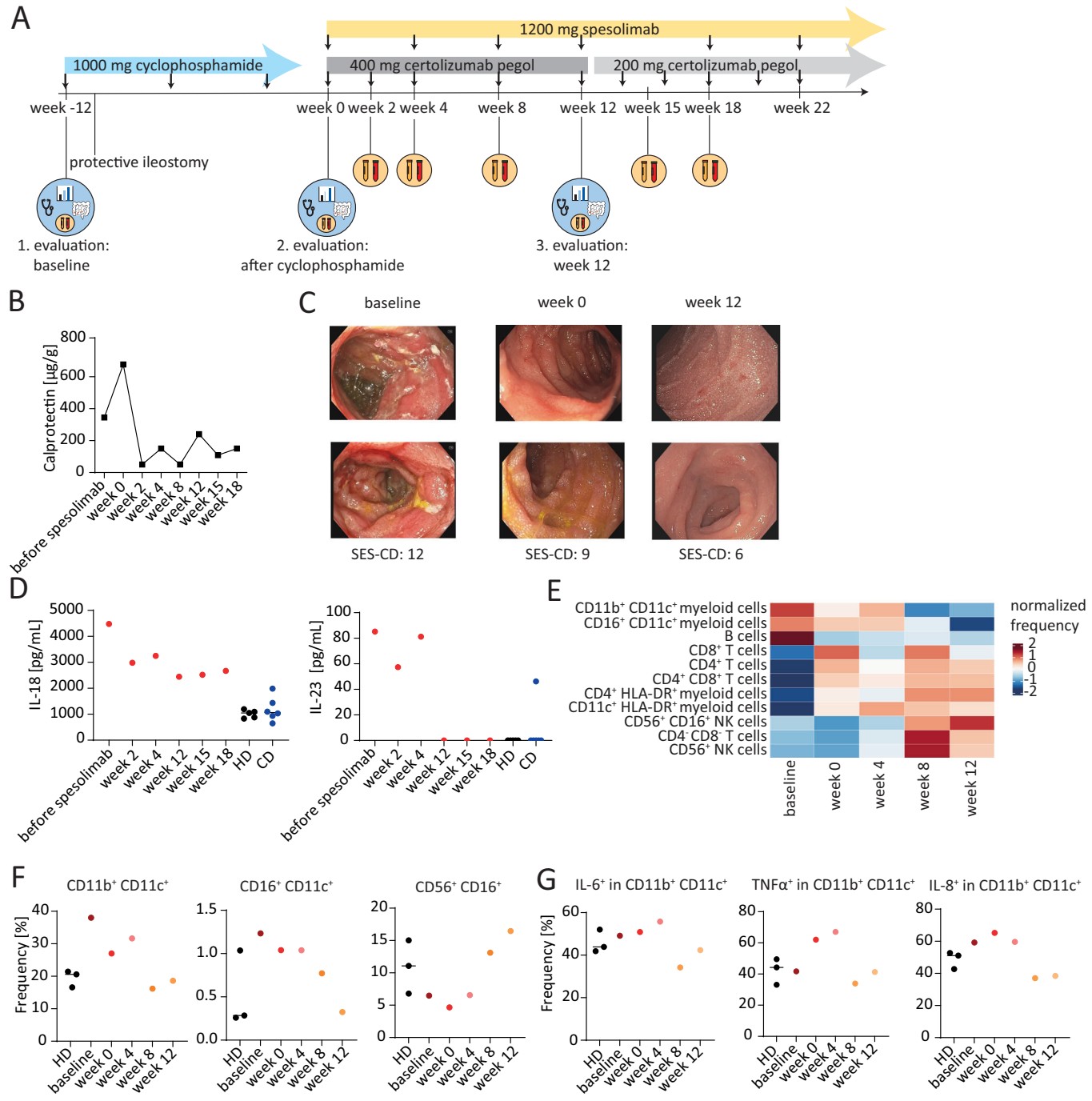

**Figure 2.  Anti-IL-36R therapy reduces intestinal inflammation in the IL-36RA patient.**

(A) The therapeutic plan of the *IL36RN*-mutated patient (IL-36RA patient). (B) Calprotectin in the stool of IL-36RA patient during spesolimab therapy. (C) Endoscopic images showing the luminal inflammation in the ileum of the IL-36RA patient before and during treatment with spesolimab and certolizumab pegol as well as the Simple Endoscopic Score for Crohn's disease (SES-CD) as assessed during colonoscopy. (D) Cytokine levels in the serum of healthy donors (HD), Crohn's disease patients (CD), and the IL-36RA patient at different time points during spesolimab therapy. The line in the plots indicates the median. HD: $n = 5$, CD: $n = 6$, IL-36RA patient: $n = 1$. (E–G) PBMCs from the IL-36RA patient before and during spesolimab therapy and PBMCs of HDs were stimulated in vitro with phorbol 12-myristate 13-acetate (PMA)/ ionomycin (Iono) or lipopolysaccharide (LPS) for 4 h or with IL-36α for 7 h. The cells were subsequently analyzed by mass cytometry. HD: $n = 3$, IL-36RA patient: $n = 1$. (E) Heatmap showing the frequency of the 11 identified clusters in unstimulated PBMCs. (F) Frequency of selected clusters in unstimulated PBMCs. (G) Frequency of pro-inflammatory cytokine-producing myeloid cells in IL-36α-stimulated samples. The line in the plots indicates the median. Source data are available online for this figure.

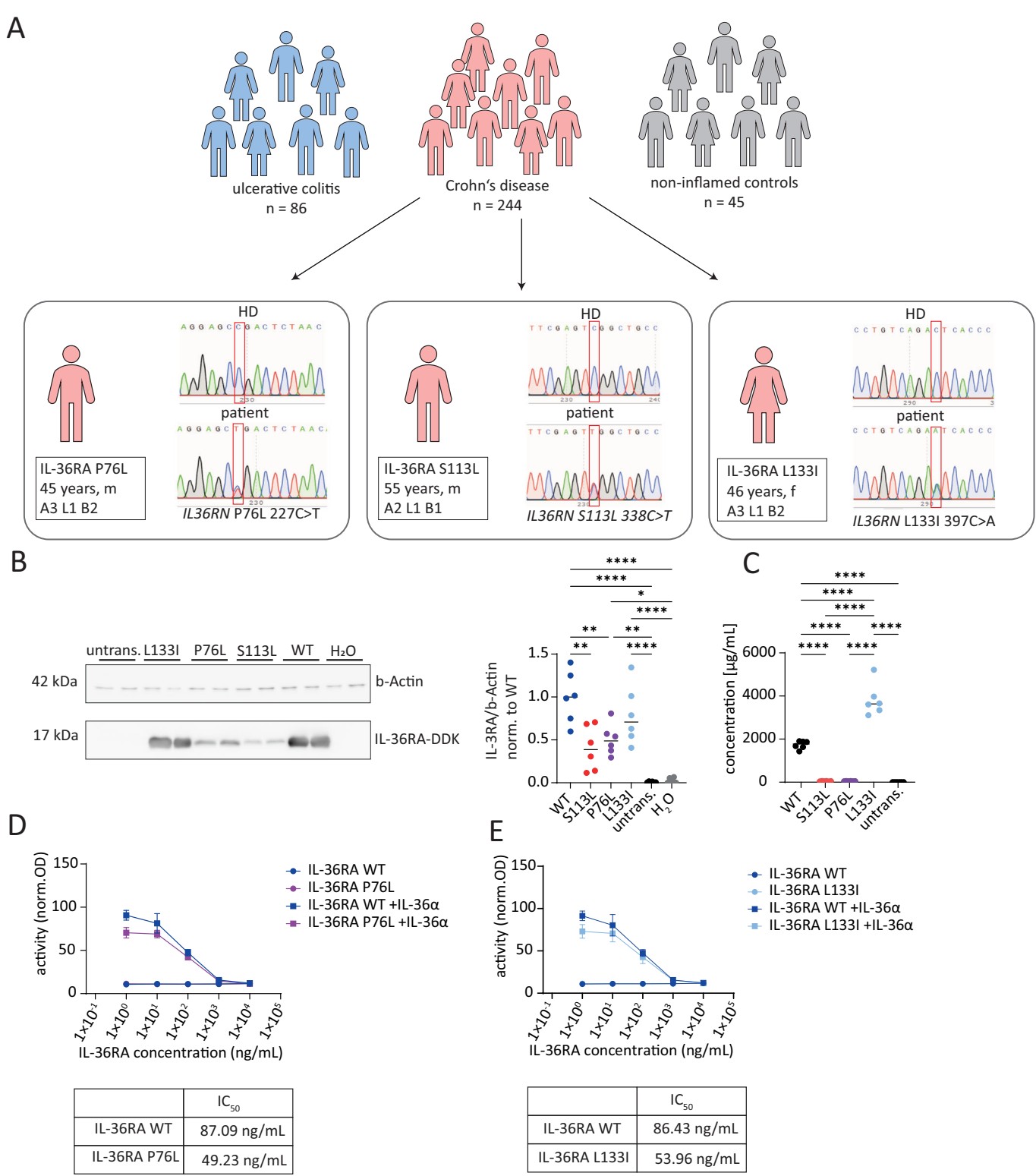

We also observed a reduction of CD8+ CD38+ T cells in LPMCs of intestinal biopsies of the *IL36RN*-mutated patient during spesolimab treatment. These observed differences may contribute to the clinical response of the *IL36RN*-mutated patient to spesolimab treatment. However, as the patient had received cyclophosphamide prior to combined treatment with spesolimab and certolizumab

pegol, it is possible that the here reported molecular changes and the clinical outcome were at least partly influenced by previous treatment with cyclophosphamide or by the combination of all three advanced therapies.

Furthermore, it was not possible to assess the long-term efficacy of spesolimab treatment, as the patient developed a small perianal

Figure 3. Pathogenic *IL36RN* mutations are present in other Crohn's disease patients.

(A) Mutations in *IL36RN* in a whole-exome sequencing dataset of 45 healthy donors, 86 patients with ulcerative colitis, and 244 patients with Crohn's disease were identified by searching for mutations predicted by PolyPhen-2 to be damaging. Subsequently, mutations were confirmed by targeted Sanger sequencing. The red square indicates the position of the mutation in the *IL36RN* gene. (B, C) HEK 293T cells were either transfected with *IL36RN* wild-type (WT), *IL36RN* P76L (P76L), *IL36RN* L133I (L133I), water ($H_2O$), or left untransfected (untrans.). IL-36RA protein expression was analyzed 48 h after transfection (B) in cell lysates by Western blot and (C) in the supernatant by ELISA. Data represent three independent experiments with $n = 6$ per condition. The line in the plots indicates the median. (D, E) NFκB activity of HEK-Blue IL-36 cells pre-incubated with different concentrations of IL-36RA WT, IL-36RA P76L, or IL-36RA L133I and subsequently stimulated with IL-36α. Values are normalized to control samples stimulated with IL-36α only. The half maximal inhibitory concentration ($IC_{50}$) for IL-36RA WT, IL-36RA P76L, and IL-36RA L133I is indicated in the table below the graph. The data represent three independent experiments. Data are represented as mean ± SD ($n = 10$). Statistical significance in (B, C) was determined by one-way ANOVA with Tukey's multiple comparisons test. ****$P < 0.0001$, ***$P < 0.001$, **$P < 0.01$, *$P < 0.05$. Exact $P$ values for the statistical comparisons are shown in Appendix Table S3. Source data are available online for this figure.

abscess at week 22, which was most likely caused by residual disease activity. Although a seton drainage was effective in draining the abscess and no increase in systemic or intestinal inflammation was observed, limited clinical data on spesolimab in the treatment of CD patients led to the decision to discontinue spesolimab treatment and monotherapy with certolizumab pegol was continued. Following the discontinuation of spesolimab, the patient experienced a severe flare requiring a second round of cyclophosphamide treatment four months after stopping the treatment with spesolimab. Based on the observed differences in IL-23 serum levels during spesolimab therapy and newly available p19 blockers, we decided to subsequently treat our patient with the anti-IL-23 antibody risankizumab. This resulted in a partial clinical response 6 months after therapy induction.

It is noteworthy that all identified patients exhibited heterozygous mutations in *IL36RN*. Autosomal-dominant immunodeficiencies frequently demonstrate incomplete clinical penetrance, suggesting that additional factors, such as environmental or infectious stimuli, may play a role in the onset of these diseases (Hadjadj et al, 2020). A comparable mechanism may also be implicated in the pathogenesis of CD in *IL36RN*-mutated patients. The combination of the *IL36RN* mutation with environmental or microbial factors may result in the development of the disease phenotype. This is also underlined by the fact that all identified *IL36RN*-mutated patients developed CD in adulthood and not in early childhood, as usually seen in monogenic IBD. This may also explain the partial clinical response of the *IL36RN*-mutated patient to spesolimab treatment, suggesting a complex disease driven by multiple factors. While our data indicate that the IL-36 pathway plays a significant role in the pathogenesis of CD in this patient, it is highly likely that additional pathways, not targeted by spesolimab therapy, are also involved in the complex therapy-refractory phenotype of our patient. Moreover, recent findings suggest that therapeutic pressure can alter immune cell composition and function, potentially leading to molecular resistance against biological therapies, including TNF blockers (Atreya and Neurath, 2022). A similar mechanism may have occurred in our *IL36RN*-mutated patient in response to spesolimab and certolizumab treatment.

Whether different *IL36RN* mutations identified in this study are associated with different severities of inflammation cannot be determined based on the current cohort and should be investigated in larger studies.

In conclusion, we have here identified four patients with ileal Crohn's disease with mutations in *IL36RN*, suggesting that *IL36RN* mutations might define a small sub-group of patients, who could benefit from anti-IL36R therapy. However, further analyses in larger cohorts will be necessary to validate our findings.

# Methods

## Reagents and tools table

| Reagent/resource | Reference or source | Identifier or catalog number |
|---|---|---|
| **Experimental models** | | |
| Patient Samples | Charité - Universitätsmedizin Berlin and Universitätsklinikum Erlangen | |
| HEK 293 T cells | ATCC | |
| HEK-Blue™ IL-36 cells | Invivogen | hkb-hil36r |
| HEK-Blue™ Null1 cells | Invivogen | hkb-null1 |
| NEB 10-β competent *Escherichia coli* | New England BioLabs | C3019H |
| ClearColi BL21(DE3) cells | Lucigen | |
| **Recombinant DNA** | | |
| *IL36RN* (NM_173170) Human Tagged ORF Clone pCMV entry | Origene | RC211691 |
| pCMV6-Entry *IL36RN* S113L Plasmid | Addgene | 58321 |
| pET-(IL36RA) | VectorBuilder | VB220506-1124dka |
| **Antibodies** | | |
| CD45 89Y | Standard BioTools | 3089003B |
| CD45 141Pr | Standard BioTools | 3141009B |
| CD19 142Nd | Standard BioTools | 3142001B |
| CD45RA 143Nd | Standard BioTools | 3143006B |
| IL-4 144Nd | Standard BioTools | 3142002B |
| CD4 145Nd | Standard BioTools | 3145001B |
| TNFα 146Nd | Standard BioTools | 3146010B |
| CD11c 147Sm | Standard BioTools | 3147008B |
| CD16 148Nd | Standard BioTools | 3148004B |
| CD25 149Sm | Standard BioTools | 3149010B |
| CD138 150Nd | Standard BioTools | 3150012B |
| IL-8 151Eu | BioLegend | 511402 |
| Fas 152Sm | Standard BioTools | 3152017B |
| IgM 153Eu | BioLegend | 314527 |
| CD3 154Sm | Standard BioTools | 3154003B |
| CD56 155Gd | Standard BioTools | 3155008B |
| IL-6 156Gd | Standard BioTools | 3156011B |
| IFNγ 158Gd | Standard BioTools | 3158017B |
| CCR7 159Tb | Standard BioTools | 3159003 A |
| CD27 160Gd | BioLegend | 302839 |
| IL-23 161Dy | Standard BioTools | 3161010B |

| Reagent/resource | Reference or source | Identifier or catalog number |
|---|---|---|
| CD8α 162Dy | Standard BioTools | 3162015B |
| CD33 163Dy | Standard BioTools | 3163023B |
| CD45RO 164Dy | Standard BioTools | 3164007B |
| CD40 165Ho | Standard BioTools | 3165005B |
| IL-2 166Er | Standard BioTools | 3166002B |
| CD38 167Er | Standard BioTools | 3167001B |
| CD40L 168Er | Standard BioTools | 3168006B |
| IL-13 169Tm | Standard BioTools | 3169016B |
| IL-12 170Er | Miltenyi | 511005 |
| CD68 171Yb | Standard BioTools | 3171011B |
| IL-17 172Yb | Standard BioTools | 3172020B |
| HLA-DR 173Yb | Standard BioTools | 3173005B |
| PD-1 174Yb | Standard BioTools | 3174020B |
| CD14 175Lu | Standard BioTools | 3175015B |
| IL-7R 176Yb | Standard BioTools | 3176004B |
| CD11b 209Bi | Standard BioTools | 3209003B |
| DYKDDDDK Tag | Cell Signaling | 14793 |
| β-actin | Sigma-Aldrich | A5441 |
| Goat anti-rabbit | Agilent | P0448 |
| Rabbit anti-mouse | Agilent | P0161 |
| CD45 BV785 | BioLegend | 304048 |
| CD19 BV750 | BioLegend | 302262 |
| CD25 BV650 | BioLegend | 302634 |
| IgM BV605 | BioLegend | 314524 |
| HLA-DR BV570 | BioLegend | 307638 |
| CD56 (NCAM) BV510 | BioLegend | 318340 |
| CD11b AF647 | BioLegend | 101218 |
| CD14 PerCP | BioLegend | 301848 |
| CD16 PE/Cy5 | BioLegend | 302010 |
| CD11c PE/Dazzle594 | BioLegend | 301642 |
| CD8 BUV563 | BD Biosciences | 612914 |
| CD3 BUV496 | BD Biosciences | 612940 |
| CD4 BUV395 | BD Biosciences | 564724 |
| γδ TCR BUV805 | BD Biosciences | 748532 |
| CD45RA BUV661 | BD Biosciences | 741654 |
| CD197 (CCR7) BUV737 | BD Biosciences | 741786 |
| CD38 BUV615 | BD Biosciences | 751138 |
| T-bet BV421 | BioLegend | 644816 |
| FOXP3 AF700 | ThermoFisher Scientific | 56-4776-41 |
| Gata-3 AF488 | ThermoFisher Scientific | 53-9966-42 |
| EOMES PE/Cy7 | ThermoFisher Scientific | 25-4877-42 |
| Rorγt PE | BD Biosciences | 563081 |
| CD14 BV421 | BioLegend | 301830 |
| CD19 FITC | eBioscience | 11-0199-42 |
| CD4 FITC | BD Biosciences | 555346 |
| CD8 V500 | BD Biosciences | 560775 |
| **Oligonucleotides and other sequence-based reagents** | | |
| *IL36RN* (S113L and L133I) Forward | | AGATGCTGAGCCTACTGAAG |
| *IL36RN* (S113L and L133I) Reverse | | TCTGACATCAGCACCTCCTC |
| *IL36RN*-2 (P76L) Forward | | TCATGACAGCTGCTGAGAAG |

| Reagent/resource | Reference or source | Identifier or catalog number |
|---|---|---|
| *IL36RN*-2 (P76L) Reverse | | AGCTGCCATCAACAGAATCC |
| pCMV-*IL36RN*-L133I Forward | | GCCTGTCAGAATCACCCAGCT |
| pCMV-*IL36RN*-L133I Reverse | | TGATCGGCTTCAGGCACC |
| pCMV-*IL36RN*-P76L Forward | | GGGCAGGAGCTGACTCTAACA |
| pCMV-*IL36RN*-P76L Reverse | | CACCCCACATGACAGGCA |
| pET-*IL36RN*-L133I Forward | | GCCGGTTCGTATTACCCAGCTGC |
| pET-*IL36RN*-L133I Reverse | | TGATCTGCTTCCGGAACG |
| pET-*IL36RN*-P76L Forward | | GGTCAAGAACTGACACTGACC |
| pET-*IL36RN*-P76L Reverse | | AACACCACAGCTCAGACA |
| pET-*IL36RN*-S113L Forward | | CAGCTTTGAACTGGCAGCA TATCCTGGTTG |
| pET-*IL36RN*-S113L Reverse | | CTGGTCAGACCCATATCAC |
| XL39 | | ATTAGGACAAGGCTGGTGGG |
| T7 | | TAATACGACTCACTATAGGG |
| **Chemicals, enzymes, and other reagents** | | |
| RPMI | Gibco | 11875093 |
| DMEM medium high glucose | Gibco | 41965062 |
| HBSS without calcium and magnesium | Gibco | 14170112 |
| Fetal calf serum | Sigma-Aldrich | |
| Penicillin-streptomycin | ThermoFisher Scientific | 15140122 |
| Normocin | Invivogen | ant-nr-1 |
| Zeocin | Invivogen | ant-zn-05 |
| Blasticidin | Invivogen | ant-bl-05 |
| LB agar (Lennox) | Carl Roth | X965.1 |
| Ampicillin | Sigma-Aldrich | 59349 |
| Kanamycin | ThermoFisher Scientific | 11815024 |
| β-mercaptoethanol | Sigma-Aldrich | M3148 |
| Dimethyl sulfoxide | Sigma-Aldrich | D2650 |
| Ficoll® Paque Plus | Merck | GE17-1440-03 |
| Brefeldin A | Sigma-Aldrich | B7651 |
| Ionomycin | Sigma-Aldrich | I0634 |
| Phorbol 12-myristate 13-acetate | Sigma-Aldrich | P8139 |
| Lipopolysaccharide | Sigma-Aldrich | L2630 |
| Chloroquine –diphosphate (salt) | Sigma-Aldrich | C6628 |
| HEPES buffered saline (2x) | Sigma-Aldrich | 51558-50 ML |
| CaCl₂ | Merck | 2382 |
| Pierce™ methanol-free formaldehyde | ThermoFisher Scientific | 28908 |
| Benzonase® Nuclease | Sigma-Aldrich | E1014-25KU |
| Cell-ID™ Intercalator-Ir—125 µM | Standard BioTools | 201192 A |
| Proteomic Stabilizer PROT1 | SMART TUBE Inc. | 501351691 |
| QUANTI-Blue™ solution | Invivogen | rep-qbs2 |

| Reagent/resource | Reference or source | Identifier or catalog number |
|---|---|---|
| Maxpar® Cell Staining Buffer | Standard BioTools | 201068 |
| Q5® High-Fidelity DNA Polymerase | New England BioLabs | M0491L |
| Recombinant human IL-36α/IL-1F6 protein (with carrier) | R&D Systems | 1078-IL-025 |
| RIPA buffer | Sigma-Aldrich | R0278 |
| Amersham ECL Western Blotting Detection Reagent | GE Healthcare Life Sciences | RPN2209 |
| Collagenase A | Merck | 10103578001 |
| DNase I | Merck | 260913 |
| Ulp1p SUMO protease | Produced in-house | |
| **Software** | | |
| Cytobank | Beckman Coulter | |
| R version 4.0.5 | www.R-project.org | |
| SnapGene Viewer 7.0 | SnapGene | |
| FlowJo V 10.06.1 | FlowJo | |
| GraphPad Prism 9 | GraphPad Software, | |
| FCAP Array™ software V3.0 | BD biosciences | |
| LEGENDplex™ Data Analysis Software | BioLegend | |
| **Other** | | |
| SepMate™-50 (IVD) | Stemcell | 85450 |
| Cell-ID™ 20-Plex Pd Barcoding Kit | Standard BioTools | 201060 |
| DNeasy Blood & Tissue kit | Qiagen | 69504 |
| eBioscience™ Foxp3/ Transcription Factor Staining Buffer Set | ThermoFisher Scientific | 00-5523-00 |
| EndoFree® Plasmid Purification kit | Qiagen | 12362 |
| EasySep™ Human T Cell Isolation Kit | Stemcell | 17951 |
| RNAlater® | Sigma-Aldrich | R0901-100ML |
| RNeasy Mini Kit | Qiagen | 74104 |
| Invisorb® Spin Plasmid Mini Two kit | Stratek | |
| LEGENDplex™ Human Inflammation Panel 1 (13-plex) | BioLegend | 740809 |
| NucleoSpin® Gel and PCR Clean-up kit | Macherey-Nagel | 740609.50 |
| Q5 site-directed Mutagenesis Kit | New England BioLabs | E0554 |
| Human IL-36RA/IL-1F5 DuoSet ELISA | R&D Systems | DY1275-05 |
| HisTrap™ FF Crude column | Cytiva | |
| 26/600 Superdex 75 prep grade column | Cytiva | |

## Ethical regulations

Written informed consent was obtained from all healthy volunteers and patients as approved by the institutional review board of Charité - Universitätsmedizin Berlin (EA1/200/17). All experiments involving human material were conducted in accordance with the principles set out in the WMA Declaration of Helsinki and the Department of Health and Human Services Belmont Report.

## Isolation of peripheral blood mononuclear cells

Peripheral blood mononuclear cells (PBMCs) were isolated by density-gradient centrifugation using SepMate™ PBMC Isolation tubes (Stemcell) and Ficoll-Paque™ PLUS (Merck) following the manufacturer's protocols. PBMCs were either used freshly or were frozen in fetal calf serum (FCS) supplemented with 10% dimethyl sulfoxide (DMSO) (Sigma-Aldrich).

## Isolation of lamina propria mononuclear cells

Lamina propria mononuclear cells (LPMCs) were isolated from intestinal biopsies collected during colonoscopy. Four biopsies per location were incubated in 20 mL 1 mM Diethiothreitol (DTT) solution for 15 min at 200 rpm at room temperature (RT). Following incubation, biopsies were washed in Hanks' Balanced Salt Solution (HBSS) without $Ca^{2+}$ and $Mg^{2+}$, supplemented with 5% Penicillin/Streptomycin. Biopsies were then incubated in digestion media containing RPMI 1640 supplemented with 10% FCS, 1% Penicillin/Streptomycin, 0.16 U/mL collagenase A (Merck) and 10 μg/mL DNase I (Sigma-Aldrich) for 1 h at 200 rpm at 37 °C. After enzymatic digestion, biopsies were mechanically disrupted by pipetting several times with a syringe and an 18 G blunt needle. The resulting cell suspension was filtered through a 100 μm cell strainer and washed once with HBSS containing 5% penicillin/streptomycin. Cells were separated using Percoll gradient centrifugation for 30 min at $300 \times g$ at 4 °C. LPMCs were collected from the 40 to 60% Percoll interface, washed twice in HBSS and frozen in FCS supplemented with 10% DMSO.

## Whole-exome sequencing

Whole-exome sequencing was performed as previously described (Ziegler et al, 2019). DNA was isolated from EDTA blood or from PBMCs of patients using the DNeasy Blood & Tissue Kit (Qiagen) following the manufacturer's instructions. Exome enrichment was performed using the IDT xGen Exome Research Panel v 1.0, and 2x75bp paired-end sequencing was carried out on an Illumina HiSeq 3000 sequencer. The reads were subsequently mapped against the human reference genome, converted to bam format and indexed with Samtools. PCR duplicates were removed, local realignment around InDels and base quality score recalibration were performed. Subsequently, variant calling and variant quality score recalibration was performed. Variant annotation and filtering were performed using the Alissa Interpret software (Agilent).

## Whole-exome sequencing of the TRR241 IBDome cohort

Germline mutations were called using a custom-built nextflow pipeline. Briefly, whole-exome sequencing raw reads were cleaned from residual adapter sequences and low-quality sequences using fastp v0.12.4 (Chen et al, 2018). The reads were then aligned to the reference genome (hg38) using BWA v0.7.17 (Li and Durbin, 2009). Duplicate reads were marked with sambamba v0.8.0 (Tarasov et al, 2015). Base-call quality score recalibration was performed with GATK4 v4.2.3 (Van der Auwera et al, 2013). Germline variants were called using the HaplotypeCaller program

from GATK4 and Strelka2 v2.9.10 (Kim et al, 2018). Variants that were called from both algorithms were used as high-confidence variants and annotated using the Ensembl variant effect prediction (VEP v104.3) tool (McLaren et al, 2016).

## Sanger sequencing

All mutations detected by whole-exome sequencing were subsequently confirmed by Sanger sequencing. For this, primer pairs were designed for each mutation (Reagents and Tools Table), and polymerase chain reaction (PCR) was performed on a T3000 Thermocycler (Biometra) using a Q5® High-Fidelity DNA Polymerase (New England BioLabs) and Q5® Reaction Buffer (New England BioLabs). The resulting PCR products were loaded onto a 1.5% agarose gel and electrophoresis was performed at 90 V for 80 min. Bands of the expected size were cut out from the agarose gel, purified with the NucleoSpin Gel and PCR Clean-up Kit (Macherey-Nagel), and subsequently analyzed by Sanger sequencing.

## Bulk RNA sequencing

For RNA sequencing, biopsies were fixed in RNAlater (Sigma-Aldrich) and afterward stored at −80 °C. RNA was isolated using the RNeasy Kit (Qiagen) according to the manufacturer´s instructions, and subsequently, RNA sequencing was performed. RNA sequencing samples were processed with the nf-core RNA-seq pipeline version 3.4 (Ewels et al, 2020). In brief, sequencing reads were aligned to the hg38/GRCh38 reference genome with GENCODE v33 annotations using STAR v2.7.7a (Dobin et al, 2013). Read counts and transcripts per million (TPM) were quantified using Salmon (Patro et al, 2017). Subsequently, data were analyzed in R using the edgeR package with default settings. Volcano plots and heatmaps were plotted with the EnhancedVolcano package and the pheatmap package, respectively.

## Cytometric bead array

Concentrations of cytokines in the serum of patients were measured by using the LEGENDplex™ Human Inflammation Panel 1 (BioLegend) according to the manufacturer's instructions. Data were analyzed by using the LEGENDplex™ Data Analysis Software (BioLegend).

## In vitro stimulation of PBMCs

Freshly isolated PBMCs were cultured in RPMI 1640 supplemented with 10% FCS, 1% Penicillin/Streptavidin and 1 µL/mL beta-mercaptoethanol (Sigma-Aldrich). Cells were either left unstimulated or subsequently treated for 4 h with either 20 ng/ml phorbol 12-myristate 13-acetate (PMA; Sigma-Aldrich) and 1 µg/ml ionomycin (Sigma-Aldrich) or with 100 ng/ml lipopolysaccharide (Sigma-Aldrich). In some experiments, cells were stimulated with 1 µg/mL IL-36α (R&D Systems) for 7 h. For mass cytometry experiments, 5 µg/ml Brefeldin A (Sigma-Aldrich) was added and for the last 15 min of stimulations, cells were supplemented with 25 units/ml Benzonase (Sigma-Aldrich). Subsequently, cells were fixed and frozen in Smart tube buffer (SMART TUBE Inc.) and stored at −80 °C prior to further analysis. For the analysis of secreted cytokines, no Brefeldin A was added, and supernatant was collected

after the incubation time and frozen at −20 °C. For experiments, in which spesolimab was added, freshly isolated PBMCs were first incubated with 1000 µg/mL spesolimab for 15 min and then stimulated with 100 ng/mL IL-36α for 4 h.

## Mass cytometry

### Barcoding and staining for mass cytometry

Fixed PBMCs were thawed at 37 °C, washed with Maxpar Cell Staining Buffer (Standard BioTools), and incubated with 25 units/mL Benzonase for 20 min at 37 °C. Samples were then barcoded with six different palladium isotopes using the Cell-ID 20-Plex Pd Barcoding Kit (Standard BioTools) following the manufacturer's protocol. After barcoding, samples were washed twice with Maxpar Cell Staining Buffer (Standard BioTools) and pooled afterwards. Pooled cells were washed again with Maxpar Cell Staining Buffer (Standard BioTools), and subsequently incubated with the antibody mix for cell surface staining for 30 min at 4 °C. After incubation, cells were washed twice with Maxpar Cell Staining Buffer (Standard BioTools) and then incubated with fixation/permeabilization buffer (ThermoFisher Scientific) for 60 min at 4 °C following the manufacturer's protocol. Cells were washed twice with permeabilization buffer (ThermoFisher Scientific) and subsequently incubated with the antibody mix for intracellular staining for 60 min at RT. Cells were then washed twice with permeabilization buffer (ThermoFisher Scientific) and twice with Maxpar Cell Staining Buffer (Standard BioTools). Cells were subsequently incubated overnight in 2% methanol-free formaldehyde solution (ThermoFisher Scientific). After fixation, cells were washed twice with Maxpar Cell Staining Buffer (Standard BioTools) and were incubated in iridium intercalator solution (Standard BioTools) for 60 min at RT. Cells were then washed twice with Maxpar Cell Staining Buffer (Standard BioTools) and washed with ddH$_2$O using the Laminar Wash Mini (Curiox). Cells were kept at 4 °C until CyTOF measurement.

## CyTOF measurement and data analyses

Cells were acquired on a CyTOF2 mass cytometer upgraded to Helios specifications (CyTOF2/Helios) (Standard BioTools). The instrument was tuned according to the manufacturer's instructions. EQ four-element calibration beads (Standard BioTools) were added to the sample for normalization of signal changes over the time of the measurement. Data analysis was performed as previously described by Böttcher et al (Bottcher et al, 2019). First, the resulting flow cytometry standard (FCS) files were normalized and then uploaded to Cytobank (www.cytobank.org) for gating of single, live cells and de-barcoding. Individual FCS files were compensated using the R package CATALYST (Nowicka et al, 2017). Compensated files were again uploaded to Cytobank, and reduced-dimensional (2D) t-SNE maps were generated. FCS files harboring the t-SNE data were downloaded from Cytobank and further analyzed using the R software. In some experiments, the t-SNE or UMAP maps were directly generated in R using the package CATALYST. For cluster identification, FlowSOM/ConsensusClusterPlus was used.

## Flow cytometry

Frozen PBMCs or LPMCs were thawed in a 37 °C water bath. After thawing, the cells were transferred to a preheated thawing medium

containing RPMI 1640 Medium supplemented with GlutaMAX™, 10% FCS, 1% Penicillin/Streptavidin, 50 μM 2-mercaptoethanol, and 50 U/mL DNase I (Sigma-Aldrich). The cells were then centrifuged at $350 \times g$ for 10 min at room temperature. Following centrifugation, the cells were resuspended in FACS buffer ($1\times$ PBS, 0.05% BSA, 0.01% NaN3, and 2 mM EDTA). The total cell suspension was prepared for flow cytometry analysis. Samples of PBMCs or LPMCs were stained with surface antibodies in FACS buffer for 30 min. Following the manufacturer's instructions, the cells were then fixed and permeabilized using the eBioscience™ Foxp3/Transcription Factor Staining Buffer Set (ThermoFisher Scientific). After fixation and permeabilization, the cells were stained intracellularly with transcription factor antibodies in permeabilization buffer. After the staining procedure, the cell suspensions were supplemented with Precision Count Beads (BioLegend) and analyzed on a BD FACSymphony flow cytometer (Configuration 5B 8 V 3 R 5YG 7UV). Daily quality control checks were performed using Sphero Rainbow Calibration Particles (BD Biosciences).

## Fluorescence-activated cell sorting

For Sanger sequencing of different immune cell populations, PBMCs were sorted as follows: Frozen PBMCs were thawed in a 37 °C water bath and washed in 10 mL MACS buffer (PBS supplemented with 0.5% BSA). CD3$^+$ and CD3$^-$ cells were then separated using the EasySep™ Human T Cell Isolation Kit (Stemcell), following the manufacturer's instructions. The CD3$^+$ cells were stained with an antibody cocktail containing CD4-FITC and CD8-V500 in MACS buffer, while CD3$^-$ cells were stained with CD14-BV421 and CD19-FITC in MACS buffer for 15 min on ice. After the incubation time, cells were washed once in MACS buffer and were afterward sorted using the FACSJazz™ Cell Sorter (BD). DNA was then extracted from the sorted cell populations using the DNeasy Blood & Tissue Kit (Qiagen) following the manufacturer's instructions.

## Vector design

For overexpression experiments, the pCMV6-Entry *IL36RN* plasmid (Origene) was used. pCMV6-Entry *IL36RN* S113L was a gift from Francesca Capon (Addgene plasmid #58321). For bacterial protein expression, the human *IL36RN* sequence was codon optimized for *E. coli*, the first base triplet encoding methionine was removed, and a pET-*IL36RN* plasmid was synthesized using VectorBuilder (https://en.vectorbuilder.com/). Mutations were introduced into the plasmids by using the Q5® Site-Directed Mutagenesis Kit (New England BioLabs) following the manufacturer's instructions. Briefly, primer pairs for the required nucleotide changes were designed, sequences were amplified, and transformed into NEB 10-β competent *E. coli* (New England BioLabs). On the next day, colonies were picked and cultured overnight in 3 mL LB media with required antibiotics (100 μg/mL ampicillin (pET-*IL36RN*) (Sigma-Aldrich) or 50 μg/mL kanamycin (pCMV-*IL36RN*) (ThermoFisher Scientific) at 37 °C and 170 rpm shaking. DNA was isolated with the Invisorb Spin Plasmid Mini Two Kit (Stratec) following the manufacturer's instructions and plasmid sequences were confirmed by Sanger sequencing. Colonies with the correct sequence were grown

overnight in 200 mL LB media with the required antibiotics (100 μg/mL ampicillin (pET-*IL36RN*) (Sigma-Aldrich) or 50 μg/mL kanamycin (pCMV-*IL36RN*)) at 37 °C and 200 rpm shaking and DNA was isolated using the EndoFree Plasmid Maxi Kit (Qiagen).

## Calcium phosphate transfection

HEK 293 T cells were cultured in DMEM supplemented with 10% FCS and 1% Penicillin/Streptavidin and were regularly tested for mycoplasma contamination. For transfection, $5 \times 10^5$ HEK 293 T cells per well were seeded in 2 mL medium in a six-well plate. When cells reached 80% confluency, calcium phosphate transfection was performed. Cells were transfected in the presence of 25 μM chloroquine with 2 μg plasmid DNA mixed with 16 μL CaCl$_2$ (2 M), 125 μL 2x HEPES buffered saline and volume was adjusted to 250 μL with water. After 4–6 h, medium was changed and after 48 h, medium and cells were collected to analyze protein expression.

## Western blot

Frozen cell pellets were thawed and lysed in 100 μL RIPA buffer (Sigma-Aldrich) containing protease and phosphatase inhibitors. For SDS-PAGE, lysates were incubated with Lämmli buffer for 10 min at 95 °C before loading 20 μg protein per lane onto a 15% gel. Gels were run at 16 mA per gel for 60–90 min. After SDS-PAGE, proteins were transferred to a polyvinylidene difluoride (PVDF) membrane at 250 mA for 60–90 min in a wet tank transfer system. Membranes were blocked in TBS-T + 5% milk at RT for 1 h and subsequently incubated in the primary antibody (anti-DDK antibody (Cell Signaling) 1:1000 and anti-β-actin antibody (Sigma-Aldrich) 1:2000 in TBS-T + 5% milk) at 4 °C overnight. After incubation, membranes were washed with TBS-T and then incubated in the secondary antibody mix (all secondary antibodies 1:2000 in TBS-T + 5% milk) for 1 h at RT. Membranes were washed with TBS-T and subsequently Western blot detection reagent (GE Healthcare) was added. Detection was performed using the image analyzer LAS-4000 mini (Fujifilm).

## ELISA

The concentration of IL-36RA in the supernatant of transfected cells was determined by using the IL-36RA ELISA from R&D Systems according to the manufacturer's instructions.

## Expression of IL-36RA proteins

The IL-36RA proteins (wild-type and variants) were produced using ClearColi BL21(DE3) cells (Lucigen) and Terrific Broth medium, supplemented with 100 μg/mL ampicillin. The cultures were grown at 37 °C until the OD$_{600}$ reached about 1.5. Gene expression was induced by the addition of 0.5 mM isopropyl β-D-1-thiogalactopyranoside at 17 °C. After induction, cultures were grown overnight at 17 °C. Cells were harvested by centrifugation and the pellets were stored at −70 °C. For purification, cells were resuspended in lysis buffer (50 mM Tris pH 8.0, 0.5 M NaCl, 5% glycerol), supplemented with 0.5 mM dithio-threitol (DTT), 1 mM phenylmethyl-sulfonyl fluoride and 100 μL 100 mg/mL lysozyme and 0.3 μL 850 U/μL benzonase per 100 mL total

**The paper explained**

**Problem**

Although significant progress has been made in understanding the pathogenesis of Crohn's disease and in developing targeted therapies, a substantial number of patients still do not respond adequately to existing treatments. This highlights the need to identify new therapeutic targets and markers to predict individual treatment responses. The IL-36 signaling pathway has recently been identified as a key regulator of intestinal homeostasis and inflammation. However, it remains unclear whether mutations in the IL-36R signaling pathway contribute to Crohn's disease pathogenesis and which patients may benefit from anti-IL-36R therapy.

**Results**

We identified a heterozygous missense mutation in *IL36RN* (*IL36RN* S113L) in a therapy-refractory Crohn's disease patient. Functional assays and overexpression experiments demonstrated that this mutation leads to reduced expression of IL-36RA. In-depth immune profiling revealed an enhanced response of PBMCs of this patient to IL-36 stimulation and elevated serum levels of IL-36-regulated cytokines. Consequently, treatment with the IL-36R-blocking antibody spesolimab, in combination with certolizumab pegol over 22 weeks, resulted in reduced intestinal inflammation and corresponding changes in immune cell composition and function. Finally, we identified three additional CD patients carrying missense *IL36RN* mutations in a cohort of 244 Crohn's disease patients.

**Impact**

Our study indicates that *IL36RN* mutations contribute to the pathogenesis of Crohn's disease in a subset of patients who may benefit from anti-IL-36R therapy. Furthermore, it emphasizes the importance of in-depth patient characterization to identify personalized treatment options in therapy refractory, difficult to treat patients with Crohn's disease.

volume and lysed by sonication (SONOPULS HD 2200, Bandelin Electronic GmbH & Co. KG). The extract was cleared by centrifugation at $55,000 \times g$ and supplemented with 20 mM imidazole pH 8.0. The protein was captured from the supernatant using affinity chromatography on a 5 mL HisTrap™ FF Crude column (Cytiva), equilibrated with 20 mM Tris-HCl pH 8.0, 0.5 M NaCl and 5 mM imidazole pH 8.0. The bound protein was washed with 5 CV (column volumes) 20 mM Tris-HCl pH 8.0, 0.5 M NaCl and 20 mM imidazole pH 8.0, followed by 5 CV 20 mM Tris-HCl pH 8.0, 0.5 M NaCl and 50 mM imidazole pH 8.0 to remove contaminating proteins. The protein was eluted with 20 mM Tris-HCl pH 8.0, 0.5 M NaCl and 0.25 M imidazole pH 8.0) and supplemented with 5 mM DTT. The fusion tag was cleaved off by adding 1:70 (w/w) yeast Ulp1p SUMO protease (produced in-house) while dialyzing into 20 mM Tris pH 8.0, 0.25 M NaCl, 5% glycerol and 1 mM DTT. The protein was supplemented with 20 mM imidazole pH 8.0 and reapplied onto the 5 mL HisTrap™ FF Crude column (Cytiva) as described above, collecting the flow through. The protein was supplemented with 5 mM DTT and further purified by gel filtration on a 26/600 Superdex 75 prep grade column (Cytiva) equilibrated with PBS buffer pH 7.4. The purified proteins were concentrated to >2 mg/mL, sterile filtered, flash-frozen in small aliquots with liquid nitrogen and stored at $-70\,°C$ until further use. The intact molecular mass of all purified constructs was confirmed by LC/MS TOF mass spectrometry.

## Functional test of recombinant IL-36RA in HEK-Blue™ IL-36 cells

HEK-Blue™ IL-36 cells (Invivogen) were trypsinized, counted and resuspended in DMEM supplemented with 10% FCS, 1% Penicillin/ Streptavidin, 100 μg/mL normocin and 100 μg/mL zeocin to obtain a cell solution of $2.8 \times 10^5$ cells/mL. In total, 180 μL of this cell solution was plated per well in a 96-well flat-bottom plate. For IL-36RA WT, IL-36RA S113L, IL-36RA P76L and IL-36RA L133I proteins, serial dilutions from 100 μg/mL to 10 ng/mL were performed. The cells were then pre-incubated with 20 μL of these dilutions for 15 min and subsequently stimulated with 10 ng/mL IL-36α for 18 h. After the incubation time, supernatant was collected and 20 μL of the supernatant was incubated with 180 μL QUANTI-Blue™ solution (Invivogen) for 30 min at 37 °C. Optical density (OD) was measured at 630 nm with an Infinite F50 plate reader (Tecan).

## Graphs

All Graphs were generated using the Prism 9 software (GraphPad).

## Statistics

Statistical tests were performed with the Prism 9 software (GraphPad). The statistical test used and the sample size are stated in the figure legends. Significant $P$ values are indicated as \*\*\*\*$P < 0.0001$, \*\*\*$P < 0.001$, \*\*$P < 0.01$, \*$P < 0.05$. Exact $P$ values for the statistical comparisons are shown in Appendix Table S3. Blinding was not performed during any of the analyses. No samples were excluded during any of the analyses.

## Data availability

The datasets produced in this study are available in the following database: Bulk RNA sequencing data of *IL36RN*-mutated patient: Zenodo https://doi.org/10.5281/zenodo.15189498. Bulk RNA sequencing data of control subjects: https://ibdome.org. Due to human data protection laws no human clinical or genomic datasets (whole-exome sequencing) were deposited but are available upon reasonable request.

The source data of this paper are collected in the following database record: biostudies:S-SCDT-10_1038-S44321-025-00245-z.

## Peer review information

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

## Acknowledgements

We would like to acknowledge the assistance of the BIH Cytometry Core Facility (BIH and Charité—Universitätsmedizin Berlin, Germany) and the Core Facility for Protein Production & Characterization (MDC Berlin). BS is supported by the German Research Foundation: CRC-TRR 241-B01 and Z02 (project number 375876048); CRU 5023 (project number 50474582), CRC 1449-B04 and Z02 (project number 431232613); CRC 1340-B06 (project number 372486779). CW is supported by the German Research Foundation: We 5303/3-2 (project number 317963045, CRC-TRR 241-A09 and B01 (project number 375876048) as well as by the Thyssen Foundation. ZT is supported by the Austrian Science Fund (FWF) (project number I6057).

## Author contributions

**Julia Hecker**: Conceptualization; Formal analysis; Investigation; Methodology; Writing—original draft; Writing—review and editing. **Christina Plattner**: Formal analysis; Methodology; Writing—review and editing. **Camila A Cancino**: Investigation; Methodology; Writing—review and editing. **Britt-Sabina Löscher**: Formal analysis; Methodology; Writing—review and editing. **Judith Saurenbach**: Formal analysis; Investigation; Writing—review and editing. **Marilena Letizia**: Methodology; Writing—review and editing. **Dietmar Rieder**: Formal analysis; Methodology; Writing—review and editing. **Inka Freise**: Investigation; Writing—review and editing. **Kristina Koop**: Supervision; Writing—review and editing. **Clemens Neufert**: Supervision; Writing—review

and editing. **Désirée Kunkel**: Resources; Supervision; Writing—review and editing. **Zainab Al Khatim**: Investigation; Writing—review and editing. **Lars-Arne Schaafs**: Formal analysis; Investigation; Writing—review and editing. **Anja Schütz**: Resources; Supervision; Writing—review and editing. **Christoph Becker**: Supervision; Writing—review and editing. **Raja Atreya**: Supervision; Writing—review and editing. **Zlatko Trajanoski**: Supervision; Writing—review and editing. **Andre Franke**: Supervision; Writing—review and editing. **Elena Sonnenberg**: Supervision; Writing—review and editing. **Ahmed N Hegazy**: Supervision; Writing—review and editing. **Britta Siegmund**: Conceptualization; Supervision; Funding acquisition; Methodology; Writing—original draft; Writing—review and editing. **Carl Weidinger**: Conceptualization; Supervision; Funding acquisition; Methodology; Writing—original draft; Writing—review and editing.

Source data underlying figure panels in this paper may have individual authorship assigned. Where available, figure panel/source data authorship is listed in the following database record: biostudies:S-SCDT-10_1038-S44321-025-00245-z.

## Funding

## Disclosure and competing interests statement

BS received grant support by Pfizer, served as consultant for Abbvie, BMS, Boehringer, Endpoint Health, Falk, Galapagos, Gilead, Lilly, MSD, Pfizer, Takeda (BS served as representative of the Charité) and received speaker's fees from Abbvie, BMS, CED Service GmbH, Falk, Ferring, Galapagos, Janssen, Lilly, Pfizer, Takeda (BS served as representative of the Charité). CW received grant support by Pfizer, served as a consultant for Pfizer and received speaker's fees from Falk, Ferring, Janssen. Boehringer Ingelheim was given the opportunity to review the manuscript for medical and scientific accuracy as well as intellectual property considerations in relation to the potential mention of BI substances. Boehringer Ingelheim had no role in the design, analysis or interpretation of the results in this study. Spesolimab was made available on the basis of an out-of-scope request.

# TRR241 IBDome Consortium

Kristina Koop[5], Clemens Neufert[5], Christoph Becker[5], Raja Atreya[5], Zlatko Trajanoski [2], Ahmed N Hegazy[1,3], Britta Siegmund [1,9]✉ & Carl Weidinger [1,9]✉

A full list of members and their affiliations appears in the Supplementary Information.

# Expanded View Figures

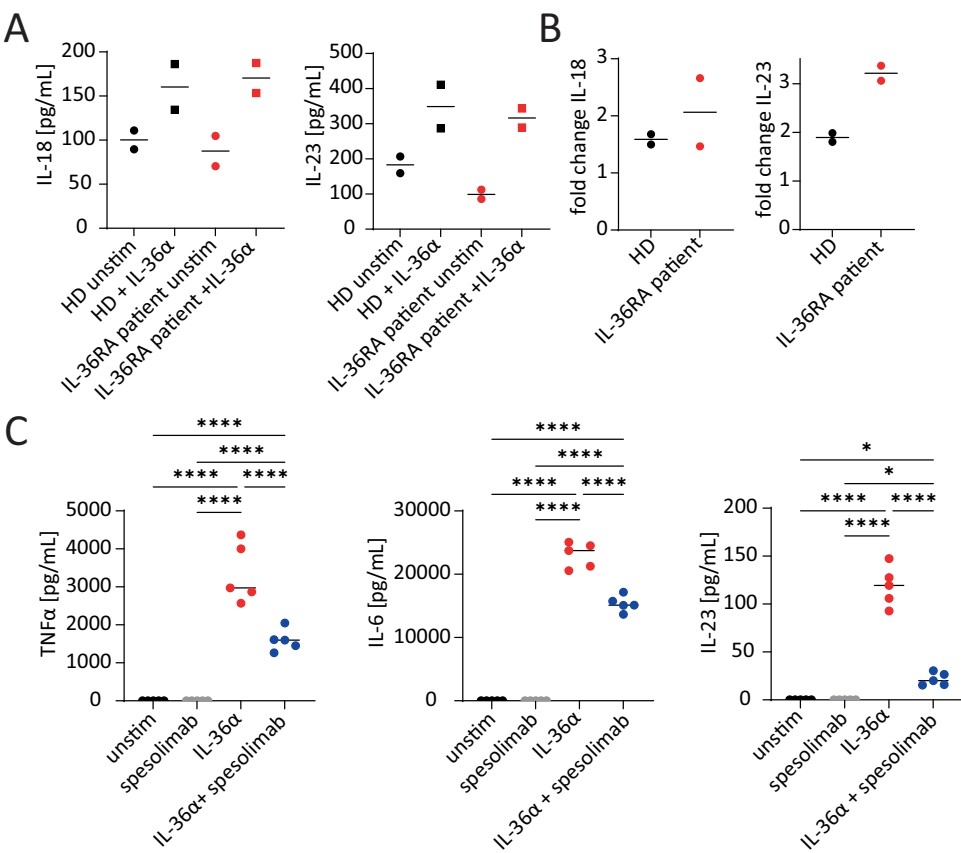

**Figure EV1. Induction of pro-inflammatory cytokines by IL-36α stimulation of PBMCs and blockade of cytokine production by the anti-IL-36R antibody spesolimab.**

(A, B) Peripheral blood mononuclear cells (PBMCs) of the *IL36RN*-mutated patient (IL-36RA patient) and one healthy donor (HD) were stimulated in vitro with IL-36α for 7 h or left unstimulated (unstim). Subsequently, cytokine levels in the supernatant were analyzed by cytometric bead array (CBA). Duplicates represent technical replicates. The line in the plots indicates the median. (B) Fold change between PBMCs stimulated with IL-36α and those left unstimulated. (C) PBMCs of the IL-36RA patient were pre-incubated with 1000 µg/mL spesolimab for 15 min and then stimulated with 100 ng/mL IL-36α for 4 h. Subsequently, the concentration of various cytokines in the supernatant was measured by CBA. Data represent technical replicates. Statistical significance was determined by one-way ANOVA with Tukey's multiple comparisons test. ****$P < 0.0001$, ***$P < 0.001$, **$P < 0.01$, *$P < 0.05$. Exact $P$ values for the statistical comparisons are shown in Appendix Table S3. Source data are available online for this figure.

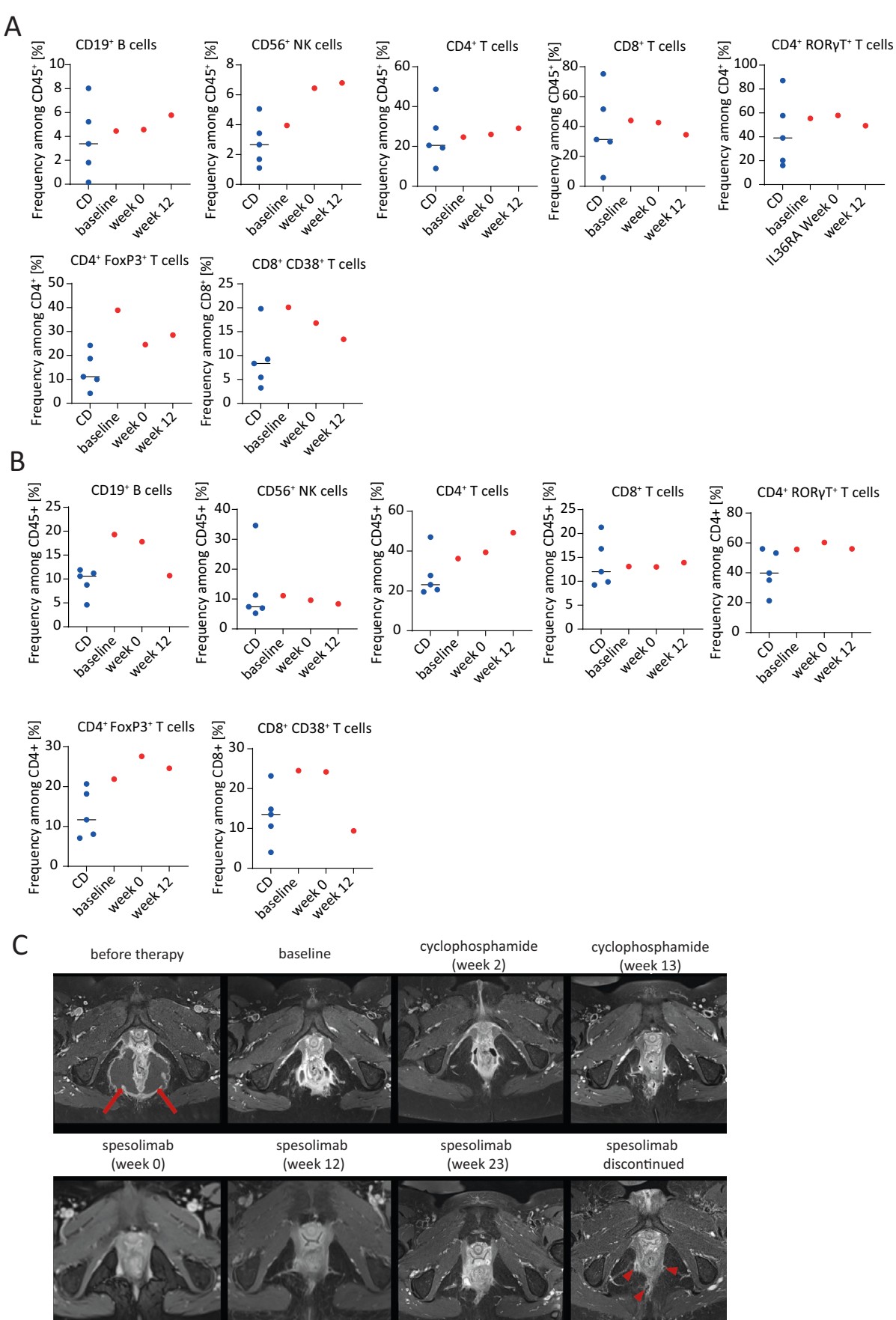

◄ **Figure EV2. Effects of spesolimab treatment on immune cell composition in the lamina propria and on abscess healing in the *IL36RN*-mutated patient.**

(A) Frequency of different immune cell populations in lamina propria mononuclear cells (LPMCs) isolated from ileal biopsies of the *IL36RN*-mutated patient (IL-36RA patient) before and during spesolimab therapy as well as of control Crohn's disease (CD) patients. CD: $n = 5$, IL-36RA patient: $n = 1$. (B) Frequency of different immune cell populations in lamina propria mononuclear cells (LPMCs) isolated from colonic biopsies of the IL-36RA patient before and during spesolimab therapy as well as of CD patients. CD: $n = 5$, IL-36RA patient: $n = 1$. (C) T1 weighted, fat-saturated MRI after i.v. contrast administration. The ischiorectal fossa is shown in an axial plane. The patient had a horseshoe perianal abscess (arrows) in January 2022, which was surgically relieved and subsequently treated with seton stitches. In the further course up to and including January 2023, the abscess healed under the sequential treatment with cyclophosphamide and spesolimab/certolizumab pegol through scarring (arrowheads). Source data are available online for this figure.

