## [Peer Review File · EMBO Molecular Medicine]

IL-36 signaling as a drug target in Crohn's disease patients with IL36RN mutations

Julia Hecker, Christina Plattner, Camila Cancino, Britt-Sabina Löscher, Judith Saurenbach, Marilena Letizia, Dietmar Rieder, Inka Freise, Kristina Koop, Clemens Neufert, Désirée Kunkel, Zainab Al Khatim, Lars-Arne Schaafs, Anja Schuetz, Christoph Becker, Raja Atreya, Zlatko Trajanoski, Andre Franke, Elena Sonnenberg, Ahmed Hegazy, Britta Siegmund, and Carl Weidinger

Corresponding authors: Carl Weidinger (carl.weidinger@charite.de) , Britta Siegmund (britta.siegmund@charite.de)

Review Timeline:

Submission Date:	4th Oct 24
Editorial Decision:	11th Nov 24
Revision Received:	1st Mar 25
Editorial Decision:	31st Mar 25
Revision Received:	17th Apr 25
Accepted:	24th Apr 25

Editor: Zeljko Durdevic

Transaction Report:

11th Nov 2024

Dear Dr. Weidinger,

Thank you for the submission of your manuscript to EMBO Molecular Medicine. We have now received feedback from the three reviewers who agreed to evaluate your manuscript. While referees #1 and #3 are overall supportive, referees #2 recognizes interest of the study but also raises important concerns that should be addressed in a major revision. If you would like to discuss further the points raised by the referees, I am available to do so via email or video. Let me know if you are interested in this option.

We would welcome the submission of a revised version within three months for further consideration. Please let us know if you require longer to complete the revision.

I look forward to receiving your revised manuscript.

Yours sincerely,

Zeljko Durdevic

We require:

- 1) A .docx formatted version of the manuscript text (including legends for main figures, EV figures and tables). Please make sure that the changes are highlighted to be clearly visible.
- 2) Individual production quality figure files as .eps, .tif, .jpg (one file per figure). For guidance, download the 'Figure Guide PDF': (<https://www.embopress.org/page/journal/17574684/authorguide#figureformat>).
- 3) A .docx formatted letter INCLUDING the reviewers' reports and your detailed point-by-point responses to their comments. As part of the EMBO Press transparent editorial process, the point-by-point response is part of the Review Process File (RPF), which will be published alongside your paper.
- 4) A complete author checklist, which you can download from our author guidelines (<https://www.embopress.org/page/journal/17574684/authorguide#submissionofrevisions>). Please insert information in the checklist that is also reflected in the manuscript. The completed author checklist will also be part of the RPF.
- 5) Please note that all corresponding authors are required to supply an ORCID ID for their name upon submission of a revised manuscript.
- 6) It is mandatory to include a 'Data Availability' section after the Materials and Methods. Before submitting your revision, primary

datasets produced in this study need to be deposited in an appropriate public database, and the accession numbers and database listed under 'Data Availability'. Please remember to provide a reviewer password if the datasets are not yet public (see <https://www.embopress.org/page/journal/17574684/authorguide#dataavailability>).

12) Author contributions: You will be asked to provide CRediT (Contributor Role Taxonomy) terms in the submission system. These replace a narrative author contribution section in the manuscript.

13) A Conflict of Interest statement should be provided in the main text.

14) Every published paper now includes a 'Synopsis' to further enhance discoverability. Synopses are displayed on the journal webpage and are freely accessible to all readers. They include a short stand first (maximum of 300 characters, including space) as well as 2-5 one-sentences bullet points that summarizes the paper. Please write the bullet points to summarize the key NEW findings. They should be designed to be complementary to the abstract - i.e. not repeat the same text. We encourage inclusion

of key acronyms and quantitative information (maximum of 30 words / bullet point). Please use the passive voice. Please attach these in a separate file or send them by email, we will incorporate them accordingly.

15) Include a Reagents and Tools Table as part of the Methods section, which can be downloaded from our author guidelines (<https://www.embopress.org/page/journal/17574684/authorguide#structuredmethods>)

**** Reviewer's comments ****

Referee #1 (Remarks for Author):

In the submitted manuscript, Hecker and colleagues report on heterogeneous mutations in the IL-36 receptor (IL36R) signaling pathway, entailing a mutated IL-36 receptor antagonist (IL36RA). The index case was identified by whole exome sequencing (WES) performed due to treatment-refractory Crohn's disease, where a mis-sense mutation in IL36RA (S113L) was noted.

In vitro experimental work identified reduced expression of IL-36RA (without detectable alterations in its function). In vitro stimulation of PBMCs from the index patient revealed elevated levels of NFkB-driven cytokines including IL-6, IL-8 and TNF- α . Concomitant alterations in PBMCs subsets were noted that included increased frequencies of TH17 cells and B cells and reduced frequency of NK cells.

Based on examination of WES data from a consortium called IBDome, where 86 UC patients, 244 CD patients and 45 controls were interrogated, 3 additional IL36RA mutations were identified, including 2 mutations in IL36RA (P76L and L33I). Further in vitro experiments were performed to characterize functional defects in IL36R signaling.

Based on these data, Spesolimab, an IL36R blocker was used to treat the index patient. This entailed partial clinical response, followed by discontinuation of the drug due to a peri rectal abscess development in the index patient. The patient was subsequently switched to a P19 inhibitor.

Rare genetic variants can be associated with immune-mediated inflammatory diseases including IBD, often with treatment-refractory phenotype. The present report demonstrates a very elegant example of 'precision biology' to treat a highly refractory patient. The investigations performed by the study team are highly complex and have an immediate clinical impact. That said, the present report also highlights the challenges in treating such individuals who often develop multiple dysregulated pathways over time. Hence early diagnosis and management of such individuals becomes extremely important.

As minor suggestions

1. The authors could consider re-organizing the presentation of their data, perhaps clubbing the in vitro work as figures 1 and 2. In the new figure 3, the authors could show the impact of treatment and perhaps show the long term effect of Spesolimab (including its discontinuation)
2. The authors could also discuss in some detail regarding the possibilities behind the lack of complete clinical response, despite a very targeted therapy in the index patient.
3. What is missing is any presentation of mucosal data. If available, it would be an invaluable addition to the manuscript. If such data are unavailable, the authors should mention this caveat.

Overall, this is a very well done study that truly defines the promise and limitations of precision-based approaches in IBD

Referee #2 (Comments on Novelty/Model System for Author):

In this study, the authors have identified four Crohn's disease patients with heterozygous missense mutations in the IL-36 receptor antagonist (IL36RN, IL-36RA) and found that two identified mutations that resulted in reduced expression of IL-36RA. In-depth immune profiling of a IL36RN patient revealed an increased response of PBMCs to IL-36 α and elevated serum levels of IL-36-regulated cytokines. Administration of the IL-36R-blocking antibody spesolimab to this patient showed a reduction

of intestinal inflammation and alterations in immune cell composition and function. However, the patient developed a small perianal abscess at week 22 of treatment cycle, and subsequently received treatment with anti-IL-23 antibody risankizumab. The topic of this study is quite interesting. However, the limitations of this study are within the following aspects and need to be addressed:

1. The authors have mentioned in the article that "Furthermore, we noted that these cytokines can be upregulated by IL-36 stimulation in vitro (Figure S3)". However, the experiment was limited to PBMCs from healthy controls (HC). It is unclear whether such changes would occur in PBMCs from IL-36RA mutant patients. It is also confused which type of immune cells is the source of this upregulation.
2. The detection of mutations in a patient's peripheral blood mononuclear cells (PBMCs) and intestinal mucosa, and whether it can be considered that the mutation could potentially be widespread among various immune cells, including T cells, B cells, monocytes, macrophages, and neutrophils?
3. It is necessary to present the evaluation of patient's (such as weight, CDAI, SES-CD, etc.) in a table during the treatment?
4. The authors evaluated the patient's SES-CD after a 12-week treatment. It is better to present the patient's endoscopic images before and after treatment for comparison?
5. Does the patient have a computer tomography angiography for the small intestine to check the mesenteric vasculature? If so, can you show the images?
6. How to explain that the serum levels of IL-23 and IL-18 in IL-36RA mutant patient before treatment were significantly higher than those in CD patients? Dose the expression of these cytokines associated with CDAI?
7. Are different mutations in IL-36R associated with the severity of inflammation in IL-36RA patient?
8. Is it possible to predict disease characteristics in patients by assessing the expression and activity levels of IL-36RA?
9. Did the author also find any changes of cytokines in IL-1b, IL-10, IL-12, IL-17, IFN-a, and TGF-b? If not, what could be the possible reasons?
10. The authors have founded that an increase in the frequency of NK cells and a decrease in the frequency of myeloid cells in PBMCs of the IL-36RA patient after treatment. Did the authors find any changes in the levels of chemokines associated with these immune cells?

Referee #2 (Remarks for Author):

This manuscript may be accepted for publication after thoroughly revised.

Referee #3 (Remarks for Author):

Hecker et al, describe the case of a patient with severe and therapy-refractory Crohn's disease. Using whole exome sequencing, the authors identified a heterozygous missense mutation in IL-36RA, which led to reduced production of IL-36RA and was associated with increased expression of inflammatory factors, and changes in immune cell populations. Notably, treatment of the patient with an anti-IL-36R blocking antibody led to partial clinical response. Further analysis of an IBD cohort revealed 3 more patients with mutations in the IL-36RA gene. These results are in accordance with previous in vitro and in vivo work showing the importance of IL-36 signaling in intestinal inflammation and repair but provide a further crucial link with human disease and the potential therapeutic utility of these studies. In addition, the identification of a target with an already established therapy could accelerate its utilization in the clinic. Studies in larger patient cohorts to establish the frequency of these mutations and address the potential utility of measuring IL-36RA levels in the clinic would be important next steps. In total, the study by Hecker et al is important in the field of IBD and in advancing personalized medicine. The manuscript is well-written, clear, and detailed, and I consider it suitable for publication in EMM in its present form.

Referee #1 (Remarks for Author):

In the submitted manuscript, Hecker and colleagues report on heterogeneous mutations in the IL-36 receptor (IL36R) signaling pathway, entailing a mutated IL-36 receptor antagonist (IL36RA). The index case was identified by whole exome sequencing (WES) performed due to treatment-refractory Crohn's disease, where a mis-sense mutation in IL36RA (S113L) was noted.

In vitro experimental work identified reduced expression of IL-36RA (without detectable alterations in its function). In vitro stimulation of PBMCs from the index patient revealed elevated levels of NFkB-driven cytokines including IL-6, IL-8 and TNF- α . Concomitant alterations in PBMCs subsets were noted that included increased frequencies of TH17 cells and B cells and reduced frequency of NK cells.

Based on examination of WES data from a consortium called IBDome, where 86 UC patients, 244 CD patients and 45 controls were interrogated, 3 additional IL36RA mutations were identified, including 2 mutations in IL36RA (P76L and L33I). Further in vitro experiments were performed to characterize functional defects in IL36R signaling.

Based on these data, Spesolimab, an IL36R blocker was used to treat the index patient. This entailed partial clinical response, followed by discontinuation of the drug due to a peri rectal abscess development in the index patient. The patient was subsequently switched to a P19 inhibitor.

Rare genetic variants can be associated with immune-mediated inflammatory diseases including IBD, often with treatment-refractory phenotype. The present report demonstrates a very elegant example of 'precision biology' to treat a highly refractory patient. The investigations performed by the study team are highly complex and have an immediate clinical impact. That said, the present report also highlights the challenges in treating such individuals who often develop multiple dysregulated pathways over time. Hence early diagnosis and management of such individuals becomes extremely important.

We thank the reviewer for his/her comments and the positive feedback on our manuscript. We have addressed the suggestions proposed by the reviewer in our revised manuscript and have included new RNA bulk and FACS analyses of intestinal biopsies obtained from the *IL36RN*-mutated index patient during spesolimab treatment as detailed below.

As minor suggestions

1. The authors could consider re-organizing the presentation of their data, perhaps clubbing the in vitro work as figures 1 and 2. In the new figure 3, the authors could show the impact of treatment and perhaps show the long term effect of Spesolimab (including its discontinuation)

We thank the reviewer for this thoughtful suggestion. However, we believe that focusing on the index patient in **Figures 1 and 2** provides a clearer understanding of the in-depth characterization and the resulting personalized treatment strategy before demonstrating the broader relevance of *IL36RN* mutations in additional CD patients in **Figure 3**. Therefore, we preferred to retain the original order of the figures. To better visualize the treatment effects of spesolimab, we have revised **Figure 2** and now include endoscopic images of the index patient, as well as sequential MRI scans throughout treatment, in the newly added **Figure EV2**, as detailed below in our response to **Reviewer 2**.

2. The authors could also discuss in some detail regarding the possibilities behind the lack of complete clinical response, despite a very targeted therapy in the index patient.

We thank the reviewer for this important comment and have addressed this concern in line 254-262 of the new discussion section of our revised manuscript. The decision to treat the index patient with spesolimab was based on the detection of the *IL36RN S113L* mutation, which has been reported in patients with generalized pustular psoriasis—a condition for which spesolimab was recently approved as a treatment (Burden et al., 2023; Onoufriadis et al., 2011).

However, we agree with the reviewer that this case represents a complex disease driven by multiple factors, which the combination treatment with spesolimab and certolizumab could only partially address. While we believe that the IL-36 pathway plays a significant role in the pathogenesis of CD in our index patient, it is highly likely that additional pathways, not targeted by spesolimab therapy, are also involved. Furthermore, recent findings suggest that therapeutic pressure can alter the composition and function of immune cells, potentially leading to the development of molecular resistance against biological therapies including TNF-blockers (Atreya et al., 2018). A similar mechanism may have occurred in our *IL36RN*-mutated patient in response to spesolimab and certolizumab treatment, contributing to the extreme therapy-refractory disease course observed in the index patient.

3. What is missing is any presentation of mucosal data. If available, it would be an invaluable addition to the manuscript. If such data are unavailable, the authors should mention this caveat.

We thank the reviewer for this insightful comment. To address this concern, we have performed bulk RNA sequencing of endoscopic intestinal biopsies from the *IL36RN*-mutated index patient before and after spesolimab therapy, as well as new FACS analyses of intestinal lamina propria mononuclear cells (LPMCs) obtained during spesolimab treatment. As shown in **Reviewer Figure 1A**, RNA sequencing analyses revealed that samples from the *IL36RN*-mutated index patient cluster distinctly from those of healthy donors (HD) and non-mutated Crohn's disease (CD) patients. Differential gene expression analyses demonstrated significant downregulation of several *HOX* family genes in the ileum of the *IL36RN*-mutated index patient compared to CD and HD samples (**Reviewer Figure 1B, 1C**), which are known to play a critical role in the embryonic development of the small and large intestine, to contribute to tissue repair, and to maintain multipotent precursor cells in adults (Bradaschia-Correa et al., 2019; Gill et al., 2024). Furthermore, in colonic samples from the *IL36RN*-mutated patient, we observed an upregulation of genes associated with antimicrobial response and tissue repair, including *REG3A*, *REG1B*, and *DEFA5*. Notably, these gene expression levels further increased under spesolimab treatment (**Reviewer Figure 1D, 1E**), suggesting that spesolimab may enhance colonic tissue repair and resilience against gut microbiota. We have included these data in **Appendix Figure S9** of our revised manuscript.

Reviewer Figure 1:

(A) MDS plot of RNA sequencing data from intestinal biopsies of the *IL36RN*-mutated patient, Crohn's disease patients (CD) and healthy donors (HD). (B) Volcano plot of RNA sequencing analyses of ileal biopsies of the *IL36RN*-mutated patient compared to CD patients (padjusted<0.05, log₂FC cutoff = 2) (C) Heatmap showing the normalized expression of selected genes in ileal biopsies (padjusted<0.05, log₂FC cutoff = 4). (D) Volcano plot of RNA sequencing analyses of colonic biopsies of the *IL36RN*-mutated patient compared to CD patients (padjusted<0.05, log₂FC cutoff = 2). (E) Heatmap showing the normalized expression of selected genes in colonic biopsies (padjusted<0.05, log₂FC cutoff = 3).

In addition, we performed flow cytometry analyses of lamina propria mononuclear cells (LPMCs) isolated of intestinal biopsies of the *IL36RN*-mutated patient during spesolimab therapy, as well as of CD patients. As shown in **Reviewer Figure 2A**, LPMCs of the ileum of the *IL36RN*-mutated patient exhibited an increased frequency of CD56⁺ natural killer (NK) cells and a reduced frequency of CD8⁺ T cells under the treatment with spesolimab. In contrast, in LPMCs of the colon, we observed a marked reduction in CD19⁺ B cells, whereas CD56⁺ NK cells and CD8⁺ T cells remained unchanged, highlighting site-specific effects of spesolimab therapy (**Reviewer Figure 2B**).

Furthermore, in LPMCs obtained from the ileum of the *IL36RN*-mutated patient, we observed a slight reduction in CD4⁺ RORγT⁺ T cells. Since RORγT is the major transcription factor of Th17 cells, this is consistent with the increased frequency of IL-17⁺ T cells in PBMCs of the *IL36RN*-mutated patient before therapy and the reduced serum levels of IL-23 during spesolimab/certolizumab-pegol therapy. Additionally, we found an increased frequency of FoxP3⁺ regulatory T cells (Tregs) in the ileum of the *IL36RN*-mutated patient, which was reduced under spesolimab treatment. This is in line with previous reports showing elevated Treg frequencies in LPMCs of IBD patients with active inflammation (Letizia et al., 2022).

In both LPMCs of the ileum and colon of the *IL36RN*-mutated patient, we observed a reduced frequency of CD8⁺ CD38⁺ T cells under spesolimab treatment, indicating that the therapy led to a

reduction in activated cytotoxic T cells, which correlates with the overall reduction in intestinal inflammation. We have included these data in **Figure EV2** of our revised manuscript.

Reviewer Figure 2:

(A) Frequency of different immune cell populations in lamina propria mononuclear cells (LPMCs) isolated from ileal biopsies of the *IL36RN*-mutated patient (IL-36RA patient) before and during spesolimab therapy as well as of Crohn's disease (CD) patients. **(B)** Frequency of different immune cell populations in lamina propria mononuclear cells (LPMCs) isolated from colonic biopsies of the IL-36RA patient before and during spesolimab therapy as well as of CD patients.

Overall, this is a very well done study that truly defines the promise and limitations of precision-based approaches in IBD

We thank the review for the kind evaluation of our paper and hope that we could improve the manuscript by the described revisions

Referee #2 (Comments on Novelty/Model System for Author):

In this study, the authors have identified four Crohn's disease patients with heterozygous missense mutations in the IL-36 receptor antagonist (IL36RN, IL-36RA) and found that two identified mutations that resulted in reduced expression of IL-36RA. In-depth immune profiling of a IL36RN patient revealed an increased response of PBMCs to IL-36alpha and elevated serum levels of IL-36-regulated

cytokines. Administration of the IL-36R-blocking antibody spesolimab to this patient showed a reduction of intestinal inflammation and alterations in immune cell composition and function. However, the patient developed a small perianal abscess at week 22 of treatment cycle, and subsequently received treatment with anti-IL-23 antibody risankizumab. The topic of this study is quite interesting.

We thank the reviewer for his/her constructive comments and hope that we could improve the manuscript according to the reviewer's suggestions. Specifically, we have added additional analyses and experiments, including measurements of chemokine levels during spesolimab treatment, performing Sanger sequencing on sorted immune cells of the *IL36RN*-mutated patient and identifying the specific cell types responding to IL-36 stimulation. Furthermore, we have evaluated endoscopic and MRI images to assess the effects of spesolimab therapy on intestinal inflammation and mesenteric vasculature as detailed below.

However, the limitations of this study are within the following aspects and need to be addressed:

1. The authors have mentioned in the article that "Furthermore, we noted that these cytokines can be upregulated by IL-36 α stimulation *in vitro* (Figure S3)". However, the experiment was limited to PBMCs from healthy controls (HC). It is unclear whether such changes would occur in PBMCs from IL-36RA mutant patients. It is also confused which type of immune cells is the source of this upregulation.

We thank the reviewer for these important comments. To address them, we have now analyzed the production of IL-23 and IL-18 in PBMCs of the *IL-36RN*-mutated patient in response to ex-vivo IL-36 α stimulation. As shown in **Reviewer Figure 3A**, PBMCs obtained from the *IL36RN*-mutated patient produced both IL-23 and IL-18 upon IL-36 α stimulation. Notably, the fold change in cytokine production, particularly IL-23, was higher in PBMCs of the *IL36RN*-mutated patient compared to those from a healthy donor (**Reviewer Figure 3B**). To investigate whether the cytokines induced by IL-36 α stimulation could be downregulated by spesolimab treatment, we furthermore stimulated PBMCs of the *IL36RN*-mutated index patient with IL-36 α *in vitro* in the presence of spesolimab. As shown in **Reviewer Figure 3C**, IL-6, TNF α , and IL-23 were upregulated upon IL-36 α stimulation and spesolimab treatment effectively reduced the concentration of these cytokines. These new analyses were included in **Figure EV1** in the revised manuscript.

Reviewer Figure 3:

(A-B) Peripheral blood mononuclear cells (PBMCs) of the *IL36RN*-mutated patient (IL-36RA patient) and one healthy donor (HD) were stimulated *in vitro* with IL-36α for 7 h or left unstimulated (unstim). Subsequently, cytokine levels in the supernatant were analyzed. Duplicates represent technical replicates. The line in the plots indicates the median. **(B)** Fold change in the respective cytokines measured in the supernatant between PBMCs stimulated with IL-36α and those left unstimulated. **(C)** PBMCs of the IL-36RA patient were pre-incubated with 1000 μg/mL spesolimab for 15 min and then stimulated with 100 ng/mL IL-36α for 4 h. Subsequently, the concentration of various cytokines in the supernatant was measured. Data represent technical replicates. Statistical significance was determined by one-way ANOVA with Tukey's multiple comparisons test. ****p<0.0001, ***p<0.001, **p<0.01, *p<0.05.

Additionally, we reanalyzed our CyTOF data to identify the cell types responsible for the pro-inflammatory cytokine response to IL-36α stimulation. As shown in **Reviewer Figure 4** we found that myeloid cells are the primary responders to IL36α stimulation in peripheral blood. We have included these data in the new **Appendix Figure S5** of the revised manuscript. However, it is important to consider that in tissue, other cell types such as macrophages, epithelial cells, and fibroblasts might also respond to IL-36 stimulation, we have therefore included an according statement in the revised discussion in line 219-222.

Reviewer Figure 4:

(A-C) Peripheral blood mononuclear cells (PBMCs) of Crohn's disease patients and healthy donors were stimulated *in vitro* with phorbol 12-myristate 13-acetate (PMA)/ionomycin (Iono) or lipopolysaccharide (LPS) for 4 h or with IL-36 α for 7 h. Subsequently, the samples were analyzed by mass cytometry. **(A)** t-SNE plots colored by the expression of selected markers. **(B)** Heatmaps showing the expression of IL-6 and TNF α in different subsets identified in PBMCs. **(C)** Mean expression of TNF α and IL-6 in CD14⁺ myeloid cells. Statistical significance was determined by paired t-test. ****p<0.0001, ***p<0.001, **p<0.01, *p<0.05.

2. The detection of mutations in a patient's peripheral blood mononuclear cells (PBMCs) and intestinal mucosa, and whether it can be considered that the mutation could potentially be widespread among various immune cells, including T cells, B cells, monocytes, macrophages, and neutrophils?

We thank the reviewer for his/her constructive comment. To address this concern, we have performed additional Sanger sequencing on sorted CD4⁺, CD8⁺, CD14⁺ and CD19⁺ cells, as well as on an intestinal biopsy from the *IL36RN*-mutated patient. The *IL36RN* S113L mutation was identified in all analyzed samples (**Reviewer Figure 5**). These findings suggest that the identified mutation is a germline mutation present in all cell types. We have added this data as **Appendix Figure S1** in the revised manuscript.

Reviewer Figure 5:

(A) Sanger sequencing of sorted CD4⁺, CD8⁺, CD14⁺ and CD19⁺ cells and of an intestinal biopsy of the *IL36RN*-mutated patient as well as of PBMCs of an unrelated control (wildtype control).

3. It is necessary to present the evaluation of patient's (such as weight, CDAI, SES-CD, etc.) in a table during the treatment?

We thank the reviewer for his/her comment. We have included a table with the weight and SES-CD of the *IL36RN*-mutated index patient during spesolimab treatment as **Appendix Table 2** in the revised manuscript. Unfortunately, the CDAI was not assessed during spesolimab treatment and can therefore not be presented here.

Reviewer Table 1:

Table showing the weight and the Simple Endoscopic Score for Crohn's Disease (SES-CD) of the *IL36RN*-mutated patient before and during treatment with spesolimab.

Timepoint	Weight [kg]	SES-CD
Baseline	Not assessed	12
Week 0 (after cyclophosphamide)	56,4	9
Week 2 of spesolimab treatment	56,4	Not assessed
Week 4 of spesolimab treatment	54,7	Not assessed
Week 8 of spesolimab treatment	56,0	Not assessed
Week 12 of spesolimab treatment	55,0	6
Week 15 of spesolimab treatment	54,0	Not assessed
Week 18 of spesolimab treatment	53,9	Not assessed

4. The authors evaluated the patient's SES-CD after a 12-week treatment. It is better to present the patient's endoscopic images before and after treatment for comparison?

We agree with the reviewer that including endoscopic images would improve the manuscript by providing visual evidence to support our findings. Therefore, we have added endoscopic images showing the luminal inflammation in the ileum before therapy initiation (baseline), after cyclophosphamide treatment (week 0) and at week 12 of spesolimab therapy. These images are included in the new **Figure 2** of the revised manuscript. Additionally, since the patient suffered from a severe penetrating disease course, we have also included MRI scans that demonstrate the resolution of the pelvic horseshoe abscess and the complete healing of the pelvic abscess under

treatment with spesolimab further underlining the clinical response to spesolimab Therapy (Appendix Figure S6 and Reviewer Figure 7).

A

Reviewer Figure 6:

(A) Endoscopic images showing the luminal inflammation in the ileum of the *IL36RN*-mutated patient before and during treatment with spesolimab.

A

Reviewer Figure 7:

(A) T1 weighted, fat-saturated MRI after i.v. contrast administration. The ischiorectal fossa is shown in an axial plane. The patient had a horseshoe perianal abscess (arrows) in January 2022, which was surgically relieved and subsequently treated with seton stitches. In the further course up to and including January 2023, the abscess healed through scarring (arrowheads).

5. Does the patient have a computer tomography angiography for the small intestine to check the mesenteric vasculature? If so, can you show the images?

We thank the reviewer for this important comment. As shown in **Reviewer Figure 8** we have now analyzed sequential abdominal and pelvine MRI scans of the *IL36RN*-mutated index patient showing

an unperturbed mesenteric vascularization under cyclophosphamide and subsequent spesolimab treatment. We included these data in the new **Appendix Figure S6** of our revised manuscript.

Reviewer Figure 8

(A) T1 weighted, fat-saturated MRI after i.v. contrast administration in the coronal plane. There is a moderately pronounced mesenteric lymphadenopathy (arrowheads) and a Comb-sign in the mid-abdomen (arrows) with otherwise unremarkable visualization of the mesenteric vascular tree over 12 months. Segmentally inflamed bowel loops were not seen at any time.

6. How to explain that the serum levels of IL-23 and IL-18 in IL-36RA mutant patient before treatment were significantly higher than those in CD patients? Dose the expression of these cytokines associated with CDAI?

Since we and others (Bridgewood et al., 2018 and Higgins et al., 2015) have shown that IL-36 α stimulation *in vitro* can induce the production of IL-23 and IL-18, the elevated levels of these cytokines in the serum of the *IL36RN*-mutated patient could be attributed to increased IL-36 signaling caused by the *IL36RN* mutation. Furthermore, we observed that IL-18 and IL-23 levels decreased with reduced inflammation in our patient, although they remained higher compared to other Crohn's disease patients. Therefore, we cannot exclude the possibility that these cytokines are also regulated by overall inflammation levels.

7. Are different mutations in IL-36R associated with the severity of inflammation in IL-36RA patient?

Since we identified only a small number of *IL36RN*-mutated patients in our IBD cohort, we unfortunately cannot determine whether different *IL36RN* mutations are associated with variations in inflammation severity. However, studies in patients with pustular psoriasis have demonstrated that different *IL36RN* mutations, as well as mutation zygosity (heterozygous, compound heterozygous, and homozygous), can influence the type and severity and inflammatory burden of the disease (Tauber et al., 2016). It is therefore in our eyes plausible to speculate that different *IL36RN* mutations may contribute to varying degrees of intestinal inflammation, which requires validation in

larger patient cohorts and investigation in future studies. We acknowledge this limitation of our study and have added an according comment in line 263-265 of the revised discussion.

8. Is it possible to predict disease characteristics in patients by assessing the expression and activity levels of IL-36RA?

We agree with the reviewer that investigating whether the expression and activity of IL-36RA could predict disease characteristics would be interesting, given that IL-36 signaling has been shown to play a role in intestinal inflammation and fibrosis (Scheibe et al., 2019). However, performing such an analysis in a timely manner of the given three months for revision would have not been feasible, as it would require the collection of a large cohort of patients. Additionally, there is currently no well-established antibody for IL-36RA staining for immunohistochemistry and no established readouts exist to assess the in-situ activity of IL-36RA.

9. Did the author also find any changes of cytokines in IL-1b, IL-10, IL-12, IL-17, IFN-a, and TGF-b? If not, what could be the possible reasons?

We thank the reviewer for his/her constructive comment. As shown in **Reviewer Figure 9** and **Appendix Figure S8**, we did not observe a clear change in IL-1 β , IL-10, IL-12, or IL-17 levels in the serum of the *IL36RN*-mutated patient during treatment with spesolimab. Within the assay used, we also measured IFN- α , but the values were below the detection limit for all samples and are therefore not included in **Appendix Figure S8**. The lack of observed differences could be explained by the low baseline levels of these cytokines in the serum of the *IL36RN*-mutated patient prior to spesolimab treatment. This might indicate that these cytokines are not major contributors to intestinal inflammation in our *IL36RN*-mutated index patient.

Reviewer Figure 9:

(A) The concentration of pro-inflammatory cytokines in the serum of healthy donors (HD), Crohn's disease patients (CD) and the *IL36RN*-mutated patient (*IL-36RA* patient) at different time points before and during spesolimab therapy.

10. The authors have found that an increase in the frequency of NK cells and a decrease in the frequency of myeloid cells in PBMCs of the *IL-36RA* patient after treatment. Did the authors find any changes in the levels of chemokines associated with these immune cells?

We thank the author for this comment and agree that changes in NK cell and myeloid cell frequency could be associated with changes in chemokine expression and migration of these cells. Therefore, we performed additional experiments and measured the expression of several chemokines in the serum of the *IL36RN*-mutated patient during spesolimab therapy (**Reviewer Figure 10**). However, we did not detect any clear changes in chemokines associated with the migration of NK cells (CXCL9, CXCL10, CXCL11, and CCL4) and myeloid cells (CCL2, CCL4, and CXCL10). Due to the limited availability of intestinal tissue from this patient, we were unfortunately not able to measure the expression of these chemokines in the intestine.

Reviewer Figure 10:

(A) The concentration of chemokines in the serum of healthy donors (HD), Crohn's disease patients (CD) and the *IL36RN*-mutated patient (IL-36RA patient) at different time points.

Referee #2 (Remarks for Author):

This manuscript may be accepted for publication after thoroughly revised.

Referee #3 (Remarks for Author):

Hecker et al, describe the case of a patient with severe and therapy-refractory Crohn's disease. Using whole exome sequencing, the authors identified a heterozygous missense mutation in *IL-36RA*, which led to reduced production of *IL-36RA* and was associated with increased expression of inflammatory factors, and changes in immune cell populations. Notably, treatment of the patient with an anti-*IL-36R* blocking antibody led to partial clinical response. Further analysis of an IBD cohort revealed 3 more patients with mutations in the *IL-36RA* gene. These results are in accordance with previous in vitro and in vivo work showing the importance of *IL-36* signaling in intestinal inflammation and repair but provide a further crucial link with human disease and the potential therapeutic utility of these

studies. In addition, the identification of a target with an already established therapy could accelerate its utilization in the clinic. Studies in larger patient cohorts to establish the frequency of these mutations and address the potential utility of measuring IL-36RA levels in the clinic would be important next steps. In total, the study by Hecker et al is important in the field of IBD and in advancing personalized medicine. The manuscript is well-written, clear, and detailed, and I consider it suitable for publication in EMM in its present form.

We thank the reviewer for his/her comments and the positive feedback on our manuscript.

31st Mar 2025

Dear Dr. Weidinger,

Thank you for the submission of your revised manuscript to EMBO Molecular Medicine. We have now received the enclosed reports from the referees that were asked to re-assess it. I am pleased to inform you that we will be able to accept your manuscript pending the following final amendments:

- 1) Authors: E-mail correspondence to Marilena Letizia could not be delivered. Please update author's e-mail address and make sure to enter correct e-mail addresses for all authors in our submission system.
- 2) Figures: We note that some panels are reused e.g. Fig. EV2C and Appendix Fig. S6A. Please cite in the respective figure legend every reused panel.
- 3) In the main manuscript file, please do the following:
 - Please address all comments suggested by our data editors listed below:
 - o Figure legends:
 1. Please note that the exact p values are not provided in the legends of figures 1C, D; 3B, C; EV1 C.
 2. Please note that in figures 1C, D; 3B, C; EV1 C there is a mismatch between the annotated p values in the figure legend and the annotated p values in the figure file that should be corrected.
 3. Please note that information related to n is missing in the legends of figures 1C, D; 2D, F, G; 3B, C; EV2 A, B.
 4. Please note that scale bar and its definition are missing for figure EV2 C.
 - Correct order and titles of manuscript sections: Abstract / Keywords / The Paper Explained / Introduction / Results / Discussion / Methods / Data Availability / Acknowledgements / Disclosure and Competing Interests Statement / References / Main Figure Legends / Tables / Expanded View Figure Legends.
 - Add up to 5 keywords.
 - Add panel callouts for Figure EV1 and EV2.
 - Rename "Competing interests" to "Disclosure Statement & Competing Interests". We updated our journal's competing interests policy in January 2022 and request authors to consider both actual and perceived competing interests. Please review the policy <https://www.embopress.org/competing-interests> and update your competing interests if necessary.
 - Merge "Disclaimer" with "Disclosure Statement & Competing Interests".
 - Author contributions: Please remove it from the manuscript and specify author contributions in our submission system. CRediT has replaced the traditional author contributions section because it offers a systematic machine-readable author contributions format that allows for more effective research assessment. You are encouraged to use the free text boxes beneath each contributing author's name to add specific details on the author's contribution. More information is available in our guide to authors: <https://www.embopress.org/page/journal/17574684/authorguide#authorshipguidelines>
 - Remove "Other contributing authors". TRR241 IBDome Consortium members should be listed in an appendix table.
 - Indicate in legends exact n and exact p values, not a range, along with the statistical test used. To keep the figures "clear" some authors found providing an Appendix table Sx with all exact p-values preferable. You are welcome to do this if you want to.
 - In Methods, statistical paragraph should reflect all information that you have filled in the Authors Checklist, especially regarding randomization, blinding, replication.
 - Remove "Ethics statements". It is sufficient to disclose these information in the Methods.
 - Please remove Reagents and Tools Table and uploaded it as a separate file. Structured Methods section includes Reagents and Tools Table followed by a Methods and Protocols section. More information on how to adhere to this format as well as downloadable templates (.docx) for the Reagents and Tools Table can be found in our author guidelines: <https://www.embopress.org/page/journal/17574684/authorguide#structuredmethods>
 - An example of a paper with Structured Methods can be found here: <https://www.embopress.org/doi/full/10.1038/s44320-024-00037-6#sec-4>
 - Raw data from large-scale datasets (WES, bulk RNA sequencing) should be deposited in one of the relevant databases and made freely available prior the publication of the manuscript. Use the following format to report the accession number of your data:

[data type]: [full name of the resource] [accession number/identifier] ([doi or URL or identifiers.org/DATABASE:ACCESSION])

Please check "Author Guidelines" for more information.

<https://www.embopress.org/page/journal/17574684/authorguide#availabilityofpublishedmaterial>

- Correct the reference citation in the text and reference list. In the text, a reference should be cited by author and year of publication. Include a space between a word and the opening parenthesis of the reference that follows. In the reference list, citations should be listed in alphabetical order. Where there are more than 10 authors on a paper, 10 will be listed, followed by "et al.". Please check "Author Guidelines" for more information.

<https://www.embopress.org/page/journal/17574684/authorguide#referencesformat>

4) Appendix: Please remove the red font and add page numbers to the table of contents. Update nomenclature of tables to

"Appendix Table S1" etc., place them after Appendix Figures and update their callouts in the main manuscript file. Please upload Appendix as a PDF file.

5) Funding: Please make sure that information about all sources of funding are complete and identical in both our submission system and in "Acknowledgments". Please correct.

6) The Paper Explained: Please remove from the synopsis file and add it to the main manuscript file.

7) Synopsis:

- Synopsis image: Please provide a visual abstract as a high-resolution jpeg file 550 px-wide x 300-600)-px high to illustrate your article.

8) As part of the EMBO Publications transparent editorial process initiative (see our Editorial at <http://embomolmed.embopress.org/content/2/9/329>), EMBO Molecular Medicine will publish online a Review Process File (RPF) to accompany accepted manuscripts. This file will be published in conjunction with your paper and will include the anonymous referee reports, your point-by-point response and all pertinent correspondence relating to the manuscript. Let us know whether you agree with the publication of the RPF and as here, if you want to remove or not any figures from it prior to publication. Please note that the Authors checklist will be published at the end of the RPF.

9) Please provide a point-by-point letter INCLUDING my comments as well as the reviewer's reports and your detailed responses (as Word file).

I look forward to reading a new revised version of your manuscript as soon as possible.

Yours sincerely,

Zeljko Durdevic

*** Instructions to submit your revised manuscript ***

1) a .docx formatted version of the manuscript text (including Figure legends and tables)

2) Separate figure files*

3) supplemental information as Expanded View and/or Appendix. Please carefully check the authors guidelines for formatting Expanded view and Appendix figures and tables at <https://www.embopress.org/page/journal/17574684/authorguide#expandedview>

4) a letter INCLUDING the reviewer's reports and your detailed responses to their comments (as Word file).

5) The paper explained: EMBO Molecular Medicine articles are accompanied by a summary of the articles to emphasize the major findings in the paper and their medical implications for the non-specialist reader. Please provide a draft summary of your

article highlighting

6) Author contributions: the contribution of every author must be detailed in a separate section.

7) EMBO Molecular Medicine now requires a complete author checklist

(<https://www.embopress.org/page/journal/17574684/authorguide>) to be submitted with all revised manuscripts. Please use the checklist as guideline for the sort of information we need WITHIN the manuscript. The checklist should only be filled with page numbers where the information can be found. This is particularly important for animal reporting, antibody dilutions (missing) and exact values and n that should be indicated instead of a range.

8) Every published paper now includes a 'Synopsis' to further enhance discoverability. Synopses are displayed on the journal webpage and are freely accessible to all readers. They include a short stand first (maximum of 300 characters, including space) as well as 2-5 one sentence bullet points that summarise the paper. Please write the bullet points to summarise the key NEW findings. They should be designed to be complementary to the abstract - i.e. not repeat the same text. We encourage inclusion of key acronyms and quantitative information (maximum of 30 words / bullet point). Please use the passive voice. Please attach these in a separate file or send them by email, we will incorporate them accordingly.

You are also welcome to suggest a striking image or visual abstract to illustrate your article. If you do please provide a jpeg file 550 px-wide x 300-600px high.

9) A Conflict of Interest statement should be provided in the main text

10) Please note that we now mandate that all corresponding authors list an ORCID digital identifier. This takes <90 seconds to complete. We encourage all authors to supply an ORCID identifier, which will be linked to their name for unambiguous name identification.

Currently, our records indicate that the ORCID for your account is 0000-0002-9948-0088.

Link Not Available

11) Include a Reagents and Tools Table as part of the Methods section, which can be downloaded from our author guidelines (<https://www.embopress.org/page/journal/17574684/authorguide#structuredmethods>)

Photos 400-800 DPI

*Additional important information regarding figures and illustrations can be found at

<https://bit.ly/EMBOPressFigurePreparationGuideline>. See also figure legend preparation guidelines:

<https://www.embopress.org/page/journal/17574684/authorguide#figureformat>

***** Reviewer's comments *****

Referee #1 (Comments on Novelty/Model System for Author):

These are novel human data with excellent technical quality

Referee #1 (Remarks for Author):

In this revised manuscript, the authors have addressed each of the Reviewers' comments through novel data or clarifications.

The manuscript is significantly improved over the previous version. No additional concerns are noted by this Reviewer, who would like to congratulate the authors on an excellent body of work.

Referee #2 (Comments on Novelty/Model System for Author):

The authors have answered all my questions, and the quality of the manuscript has markedly improved.

Referee #2 (Remarks for Author):

I would like to suggest an acceptance for publication in the EMM.

The authors addressed the remaining editorial issues.

24th Apr 2025

Dear Dr. Weidinger,

We are pleased to inform you that your manuscript is accepted for publication and is now being sent to our publisher to be included in the next available issue of EMBO Molecular Medicine.

Zeljko Durdevic
Senior Editor
EMBO Molecular Medicine
